# Botox Optimization Algorithm: A New Human-Based Metaheuristic Algorithm for Solving Optimization Problems

**DOI:** 10.3390/biomimetics9030137

**Published:** 2024-02-23

**Authors:** Marie Hubálovská, Štěpán Hubálovský, Pavel Trojovský

**Affiliations:** Department of Technics, Faculty of Education, University of Hradec Kralove, 50003 Hradec Králové, Czech Republic; marie.hubalovska@uhk.cz (M.H.); stepan.hubalovsky@uhk.cz (Š.H.)

**Keywords:** optimization, human-inspired, metaheuristic, Botox, exploration, exploitation

## Abstract

This paper introduces the Botox Optimization Algorithm (BOA), a novel metaheuristic inspired by the Botox operation mechanism. The algorithm is designed to address optimization problems, utilizing a human-based approach. Taking cues from Botox procedures, where defects are targeted and treated to enhance beauty, the BOA is formulated and mathematically modeled. Evaluation on the CEC 2017 test suite showcases the BOA’s ability to balance exploration and exploitation, delivering competitive solutions. Comparative analysis against twelve well-known metaheuristic algorithms demonstrates the BOA’s superior performance across various benchmark functions, with statistically significant advantages. Moreover, application to constrained optimization problems from the CEC 2011 test suite highlights the BOA’s effectiveness in real-world optimization tasks.

## 1. Introduction

Optimization problems, characterized by multiple feasible solutions, involve finding the best solution among them. Mathematically, these problems consist of decision variables, constraints, and an objective function. The optimization process aims to determine optimal values for decision variables, adhering to constraints while optimizing the objective function. Numerous real-world applications in science, engineering, industry, and technology necessitate effective optimization techniques. Two main approaches, deterministic and stochastic, address these challenges. Deterministic approaches, including gradient-based and non-gradient-based methods, excel in handling simpler problems but face limitations in complexity and local optima traps. To address complex, nonlinear, and high-dimensional challenges, researchers have developed stochastic approaches, acknowledging the limitations of deterministic methods in practical optimization scenarios [1,2,3,4,5,6].

Metaheuristic algorithms represent a widely employed stochastic approach for effective optimization problem-solving. Leveraging random search, random operators, and trial-and-error processes, these algorithms yield suitable solutions. The optimization process initiates with the random generation of candidate solutions, progressively enhancing them through iterations. The final output is the best-improved candidate solution. While the inherent randomness poses challenges in guaranteeing a global optimal solution, solutions obtained from metaheuristic algorithms are considered to be quasi-optimal due to their proximity to the global optimum. The pursuit of more effective quasi-optimal solutions, closely aligning with the global optimum, drives the development of various metaheuristic algorithms [7,8].

For metaheuristic algorithms to effectively address optimization problems, they must conduct thorough searches at both the global and local levels within the problem-solving space. Global search, aligned with exploration, denotes the algorithm’s proficiency in extensively exploring the problem-solving space to identify the region containing the primary optimum and avoid local optima. Local search, associated with exploitation, illustrates the algorithm’s ability to closely investigate promising solutions, aiming for convergence to the global optimal solution. The success of a metaheuristic algorithm is contingent on striking a balance between exploration and exploitation throughout the search process [9].

The central research inquiry revolves around whether, given the multitude of existing metaheuristic algorithms, there remains a necessity to develop novel ones. In addressing this query, the No Free Lunch (NFL) principle [10] asserts that a metaheuristic algorithm’s success in optimizing a specific set of problems does not guarantee comparable performance across all optimization tasks. The NFL theorem posits that no single metaheuristic algorithm can be deemed the optimal solution for all optimization challenges. It highlights the unpredictability of an algorithm’s success or failure in addressing different optimization problems, emphasizing that a method that is successful in converging to the global optimum for one problem may encounter difficulties, such as local optima entrapment, when applied to another problem. Consequently, the NFL theorem discourages assumptions about the universal effectiveness of a metaheuristic algorithm and encourages ongoing exploration and introduction of new algorithms to enhance solutions for diverse optimization problems.

This paper brings innovation and novelty to the forefront by introducing the Botox Optimization Algorithm (BOA), a novel metaheuristic approach for solving optimization problems. The key contributions of this paper encompass the following:Introducing the BOA involves emulating the Botox injection process, drawing inspiration from enhancing facial beauty by addressing defects in specific facial areas.BOA theory is described and then mathematically modeled.The BOA’s performance is rigorously assessed using the CEC 2017 test suite, showcasing its efficacy in solving optimization problems.The algorithm’s robustness is further tested in handling real-world applications, particularly in optimizing twenty-two constrained problems from the CEC 2011 test suite.The BOA’s performance is objectively compared with twelve established metaheuristic algorithms, establishing its competitive edge and effectiveness.

This paper follows a structured outline: Section 2 encompasses a comprehensive literature review. Section 3 introduces and models the Botox Optimization Algorithm. Section 4 presents simulation studies and results. The efficacy of the BOA in real-world applications is explored in Section 5. The paper concludes with Section 6, offering conclusions and suggestions for future research.

## 2. Literature Review

Metaheuristic algorithms draw inspiration from diverse sources, such as natural phenomena, living organisms’ lifestyles, the laws of physics, biology, human interactions, and game rules. Classified into five groups based on their design principles, these are swarm-based, evolutionary-based, physics-based, human-based, and game-based approaches.

Swarm-based algorithms, like Particle Swarm Optimization (PSO) [11], Ant Colony Optimization (ACO) [12], Artificial Bee Colony (ABC) [13], and the Firefly Algorithm (FA) [14], emulate the behaviors of animals, insects, plants, birds, and aquatic life. PSO models the group movement of birds or fish searching for food, ACO is inspired by ants finding the shortest communication path, ABC mimics honey bees’ activities in locating food, and the FA replicates fireflies’ optical communication. Noteworthy wildlife activities, such as foraging, hunting, chasing, migration, and digging, serve as the foundation for swarm-based metaheuristic algorithms like the Pufferfish Optimization Algorithm (POA) [15], Golden Jackal Optimization (GJO) [16], Tunicate Swarm Algorithm (TSA) [17], Coati Optimization Algorithm (COA) [18], Chameleon Swarm Algorithm (CSA) [19], Wild Geese Algorithm (WGA) [20], White Shark Optimizer (WSO) [21], Grey Wolf Optimizer (GWO) [22], African Vultures Optimization Algorithm (AVOA) [23], Mantis Search Algorithm (MSA) [24], Marine Predator Algorithm (MPA) [25], Whale Optimization Algorithm (WOA) [26], Orca Predation Algorithm (OPA) [27], Reptile Search Algorithm (RSA) [28], Honey Badger Algorithm (HBA) [29], and Kookaburra Optimization Algorithm (KOA) [30].

Evolutionary-based metaheuristic algorithms derive inspiration from the biological sciences, genetics, survival of the fittest, natural selection, and random operators. Prominent algorithms in this group include the Genetic Algorithm (GA) [31] and Differential Evolution (DE) [32], designed to emulate reproduction and Darwin’s theory of evolution, and to incorporate random operators like mutation, crossover, and selection. Artificial Immune Systems (AISs) are modeled after the human body’s defense system [33]. Other algorithms in this category encompass Genetic Programming (GP) [34], Cultural Algorithm (CA) [35], and Evolution Strategy (ES) [36].

Physics-based metaheuristic algorithms are developed by simulating laws, forces, transformations, and other concepts from physics. Simulated Annealing (SA) [37], a widely used algorithm in this category, emulates the metal annealing process, where metals are melted and slowly cooled to achieve optimal crystal formation. Various algorithms, including the Momentum Search Algorithm (MSA) [38], Spring Search Algorithm (SSA) [39], and Gravitational Search Algorithm (GSA) [40], are based on physical forces and Newton’s laws of motion. The Black Hole Algorithm (BHA) [41] and Multi-Verse Optimizer (MVO) [42] draw inspiration from cosmological concepts. Other physics-based metaheuristic algorithms include the Equilibrium Optimizer (EO) [43], Archimedes Optimization Algorithm (AOA) [44], Henry Gas Optimization (HGO) [45], Electro-Magnetism Optimization (EMO) [46], Lichtenberg Algorithm (LA) [47], Nuclear Reaction Optimization (NRO) [48], Thermal Exchange Optimization (TEO) [49], and Water Cycle Algorithm (WCA) [50].

Human-based metaheuristic algorithms are designed to emulate human behaviors, interactions, thoughts, and social activities. Notably, Teaching–Learning-Based Optimization (TLBO) draws inspiration from educational interactions in classrooms, simulating knowledge exchange among teachers and students [51]. The Special Forces Algorithm (SFA) mirrors real-life special forces missions, incorporating mechanisms to simulate UAV-assisted searches and contact loss due to force majeure [52]. The Political algorithm (PO) [53] replicates democratic parliamentary politics, offering a unique optimization approach inspired by political decision-making dynamics. The Chef-Based Optimization Algorithm (CHBO) [54] takes cues from individuals learning cooking skills in classes. Other human-based metaheuristic algorithms include the Coronavirus Herd Immunity Optimizer (CHIO) [55], Doctor and Patient Optimization (DPO) [56], War Strategy Optimization (WSO) [57], Election-Based Optimization Algorithm (EBOA) [58], Gaining Sharing Knowledge-Based Algorithm (GSK) [59], Following Optimization Algorithm (FOA) [60], Driving Training-Based Optimization (DTBO) [5], Sewing Training-Based Optimization (STBO) [61], and Ali Baba and the Forty Thieves (AFT) [62].

Game-based metaheuristic algorithms are formulated by simulating player behavior, influential figures, and the rules of various individual and team games. Algorithms like Football Game-Based Optimization (FGBO) [63] and Volleyball Premier League (VPL) [64] are inspired by modeling league matches. The Hide Object Game Optimizer (HOGO) [65] is designed based on players’ attempts to locate hidden objects on the playing field. The Darts Game Optimizer (DGO) [66] incorporates the skill of players throwing darts to earn more points. The Orientation Search Algorithm (OSA) [67] emulates players’ movements directed by referees. Other game-based metaheuristic algorithms include the Dice Game Optimizer (DGO) [68], Golf Optimization Algorithm (GOA), League Championship Algorithm (LCA) [6], Ring Toss Game-Based Optimization (RTGBO) [69], and Puzzle Optimization Algorithm (POA) [70].

In addition to the original versions of metaheuristic algorithms, many researchers have tried to improve the performance of existing algorithms by developing their improved versions, such as the Enhanced Snake Optimizer (ESO) [71], Improved Sparrow Search Algorithm (ISSA) [72], and multi-strategy-based Adaptive Sine–Cosine Algorithm (ASCA) [73].

To the best of our knowledge, as gleaned from the literature review, no metaheuristic algorithm inspired by the human activity of Botox injections has been introduced thus far. The process of enhancing facial beauty by injecting substances to eliminate facial defects presents an intelligent methodology that could serve as the foundation for a novel metaheuristic algorithm. To bridge this research gap in metaheuristic algorithm studies, this paper introduces a new human-based metaheuristic algorithm, grounded in the mathematical modeling of Botox injections in specific facial areas, as elaborated in the subsequent section.

## 3. Botox Optimization Algorithm

Within this section, the Botox Optimization Algorithm (BOA) is elucidated, beginning with an exploration of its theory and source of inspiration. Following this, the mathematical modeling of the implementation steps for the proposed BOA approach is detailed.

### 3.1. Inspiration of BOA

Enhancing facial beauty is a significant and intricate concern for many individuals, with the emergence of facial wrinkles often causing distress. Wrinkles result from the repetitive contraction of underlying facial muscles and dermal atrophy. To address this issue, small doses of botulinum toxin are strategically injected into specific overactive muscles. This injection induces localized muscle relaxation, subsequently leading to the smoothing of the skin in these hyperactive muscle areas [74]. Botulinum toxin, a potent neurotoxin protein derived from the bacterium Clostridium botulinum, is employed for this purpose. The administration of this toxin results in the targeted muscles being temporarily paralyzed, preventing the formation of wrinkles in the treated area [75]. Botox, the cosmetic use of botulinum toxin, gained approval from the U.S. Food and Drug Administration (FDA) in 2002 for treating glabellar complex muscles responsible for frown lines, and in 2013 for addressing lateral orbicularis oculi muscles associated with crow’s feet [76].

Botox exerts a significant impact on diminishing facial wrinkles and enhancing facial aesthetics. The strategic injection of Botox into specific facial areas to eliminate wrinkles serves as an intelligent process, forming the foundational concept behind the design of the approach proposed by the BOA.

### 3.2. Algorithm Initialization

The proposed BOA methodology operates as a population-based optimizer, leveraging the collective search capabilities of its participants in an iterative process to generate viable solutions for optimization problems. In this context, individuals seeking Botox injections constitute the BOA population. Each member contributes to decision variable values based on their position in the problem-solving space, mathematically represented as a vector. This vector, encapsulating decision variables, forms the population matrix outlined in Equation (1); initialization of each BOA member’s position is achieved through random assignment using Equation (2):(1)$$X={\left[\begin{array}{c} {\vec{X}}_{1}\\ \vdots \\ {\vec{X}}_{i}\\ \vdots \\ {\vec{X}}_{N} \end{array}\right]}_{N\times m}={\left[\begin{array}{ccccc} x_{1,1} & \cdots & x_{1,d} & \cdots & x_{1,m}\\ \vdots & \ddots & \vdots & ⋰ & \vdots \\ x_{i,1} & \cdots & x_{i,d} & \cdots & x_{i,m}\\ \vdots & ⋰ & \vdots & \ddots & \vdots \\ x_{N,1} & \cdots & x_{N,d} & \cdots & x_{N,m} \end{array}\right]}_{N\times m},$$
(2)$$x_{i,d}=lb_{d}+r_{i,d}\cdot (ub_{d}-lb_{d}),\;i=1,\ldots ,N,\;d=1,\ldots ,m,$$
where X is the BOA population matrix, ${\vec{X}}_{i}$ is the ith BOA member (candidate solution), $x_{i,d}$ is its dth dimension in the search space (decision variable), N is the number of population members, m is the number of decision variables, $r_{i,d}$ are random numbers from interval $\left[0,\;1\right]$, and $lb_{d}$ and $ub_{d}$ are the lower bound and upper bound of the dth decision variable, respectively.

Given that each member in the BOA population represents a candidate solution for the problem, the associated objective function of the problem can be assessed for each individual. Consequently, the array of objective function values can be depicted as a vector, as per Equation (3):(3)$$\vec{F}={\left[\begin{array}{c} F_{1}\\ \vdots \\ F_{i}\\ \vdots \\ F_{N} \end{array}\right]}_{N\times 1}={\left[\begin{array}{c} F({\vec{X}}_{1})\\ \vdots \\ F({\vec{X}}_{i})\\ \vdots \\ F({\vec{X}}_{N}) \end{array}\right]}_{N\times 1},$$
where $\vec{F}$ is the vector of the evaluated objective function and $F_{i}$ is the evaluated objective function based on the ith BOA member.

The assessed objective function values serve as reliable criteria for appraising the quality of candidate solutions. Consequently, the optimal member of the BOA corresponds to the best value achieved for the objective function, while the suboptimal member aligns with the worst value. Given that the position of BOA population members and their objective function values are updated in each iteration, the best candidate solution undergoes regular updates.

### 3.3. Mathematical Modeling of BOA

The BOA approach, a population-based optimizer, adeptly furnishes viable solutions for optimization problems through an iterative process. In the BOA’s design, inspiration is drawn from the Botox injection mechanism to update the position of population members within the search space. The schematic of Botox injection and its simulation to design the proposed BOA approach is shown in Figure 1.

Each individual seeking Botox injections represents a member of the BOA population. The BOA design mirrors the process of a doctor injecting Botox into specific facial muscles to diminish wrinkles and enhance beauty. Similarly, in the BOA approach, improvement to a candidate solution involves adding a designated value, akin to Botox, to select decision variables.

In the design of the BOA, it is considered that the number of facial muscles that need to be injected with Botox decreases during the iterations of the algorithm. Therefore, the number of selected muscles (i.e., decision variables) for Botox injection is determined by using Equation (4):(4)$$N_{b}=\left\lfloor 1+\frac{m}{t}\right\rfloor \leq m,$$
where $N_{b}$ is the number of muscles requiring Botox injection and t is the current value of the iteration counter.

When the applicant visits the doctor, the doctor decides which muscles to inject Botox into, based on the person’s face and wrinkles. Inspired by this fact, in BOA design, the variables to be injected are selected for each population member using Equation (5). It should be noted that the muscles that are chosen for Botox injection should not be repeated, which is considered in Equation (5):(5)$$CBS_{i}=\left\{d_{1},\;d_{2},\;\ldots ,d_{j},\;\ldots ,d_{N_{b}}\;\right\},\;d_{j}\in \left\{1,2,\;\ldots ,m\right\}\;\;and\;\forall \;h,k\in \left\{1,2,\;\ldots ,N_{b}\right\}:d_{h}\neq d_{k}.$$

Thus, $CBS_{i}$ is the set of candidate decision variables of the ith population member that are selected for Botox injection, and $d_{j}$ is the position of the jth decision variable selected for Botox injection.

In the BOA design, akin to the doctor’s discretion in determining the drug quantity for Botox injection based on expertise and patient needs, the amount of Botox injection for each population member is computed using Equation (6):(6)$${\vec{B}}_{i}=\left\{\begin{array}{c} {\vec{X}}_{mean}-{\vec{X}}_{i},\;t<\frac{T}{2}\;;\\ {\vec{X}}_{best}-{\vec{X}}_{i},\;else, \end{array}\right.$$
where ${\vec{B}}_{i}=({b_{i,1},\ldots ,b}_{i,j},\ldots ,b_{i,m})$ is the considered amount for Botox injection to the ith member, ${\vec{X}}_{mean}$ is the mean population position (i.e., ${\vec{X}}_{mean}=\frac{1}{N}\sum_{i=1}^{N}{\vec{X}}_{i}$), T is the total number of iterations, and ${\vec{X}}_{best}$ is the best population member.

After Botox injection into the facial muscles, the appearance of the face changes, with the disappearance of wrinkles. In the BOA design, based on the simulation of Botox injection to the facial muscles, first, a new position is calculated for each BOA member based on Botox injection using Equation (7); then, if the value of the objective function is improved, this new position replaces the previous position of the corresponding member according to Equation (8):(7)$${{\vec{X}}_{i}^{new}:\;x}_{i,d_{j}}^{new}=x_{i,d_{j}}+r_{i,d_{j}}\cdot b_{i,d_{j}},$$
(8)$${\vec{X}}_{i}=\left\{\begin{array}{c} {\vec{X}}_{i}^{new},\;F_{i}^{new}<F_{i}\\ {\vec{X}}_{i},\;\mathrm{else}, \end{array}\right.$$
where ${\vec{X}}_{i}^{new}$ is the new position of the ith BOA member after Botox injection, ${\;x}_{i,d_{j}}^{new}$ is its $d_{j}$th dimension, $F_{i}^{new}$ is its objective function value, $r_{i,d_{j}}$ is a random number with a uniform distribution on the interval $\left[0,\;1\right]$, and $b_{i,d_{j}}$ is the $d_{j}$th dimension of Botox injection for the ith BOA member (i.e., ${\vec{B}}_{i}$).

### 3.4. Repetition Process, Pseudocode, and Flowchart of the BOA

After updating the position of all BOA members in the search space, the first iteration of the algorithm is completed. Then, based on the updated values, the algorithm enters the next iteration, and the process of updating the BOA population members continues until the last iteration, based on Equations (4)–(8). In each iteration, the best obtained candidate solution ${\vec{X}}_{best}$ is also updated and saved. After the full implementation of the proposed BOA approach, the best candidate solution ${\vec{X}}_{best}$ stored during the iterations of the algorithm is introduced as the solution to the given problem. The steps of BOA implementation are presented in the form of a flowchart in Figure 2, and its pseudocode is shown in Algorithm 1.
**Algorithm 1.** Pseudocode of the BOA.Start the BOA.1.Input problem information: variables, objective function, and constraints.2.Set the BOA population size *N* and the total number of iterations T.3.Generate the initial population matrix at random using Equation (2). 4.Evaluate the objective function.5.Determine the best candidate solution ${\vec{X}}_{best}$.6.For t=1 to *T*7.  Update number of decision variables for Botox injections using Equation (4). 8.  For i=1 to N9. Determine the variables that are considered for Botox injection using Equation (5).10. Calculate the amount of Botox injection using Equation (6). 11. For j=1 to $N_{b}$12. Calculate the new position of the ith BOA member using Equation (7). 13. End14. Evaluate the objective function based on ${\vec{X}}_{i}^{new}$.15. Update the ith BOA member using Equation (8). 16.  End17.  Save the best candidate solution obtained so far.18.  End 19.  Output the best quasi-optimal solution obtained with the BOA.End the BOA.

### 3.5. Computational Complexity of the BOA

In this subsection, the computational complexity of the BOA is evaluated. The preparation and initialization steps of the BOA for an optimization problem have a computational complexity equal to O(Nm), where N is the number of population members and m is the number of decision variables of the problem. In each iteration, the position of the population members is updated and the corresponding objective function is also evaluated. Therefore, the BOA update process has a computational complexity equal to O(NmT), where *T* is the maximum number of iterations of the algorithm. According to this, the total computational complexity of the proposed BOA approach is equal to O(Nm(1+T)).

### 3.6. Population Diversity, Exploration, and Exploitation Analysis

The population diversity of the BOA refers to the distribution of population members within the problem space, which plays a critical role in monitoring the search processes of the algorithm. Essentially, this metric indicates whether the population members are focused on exploration or exploitation. By measuring the diversity of the BOA population, it becomes possible to gauge and adapt the algorithm’s capacity to explore and exploit a collective group effectively. Various definitions of diversity have been put forth by researchers. Pant [77] defined diversity according to Equations (9) and (10):(9)$$Diversity=\frac{1}{N}\sum_{i=1}^{N}\sqrt{{\sum_{d=1}^{m}\left(x_{i,d}-{\bar{x}}_{d}\right)}^{2}},$$
(10)$${\bar{x}}_{d}=\frac{1}{N}\sum_{i=1}^{N}x_{i,d}$$
where N is the number of population members, m is the number of problem dimensions, and ${\bar{x}}_{d}$ is the mean of the entire population in the dth dimension. Hence, the percentage of exploration and exploitation of the population for each iteration can be defined by Equations (11) and (12), respectively:(11)$$Exploration=\frac{Diversity}{Diversity_{max}},$$
(12)Exploitation=1−Exploration.

In this subsection, the analysis of population diversity, exploration, and exploitation is evaluated on twenty-three standard benchmark functions, consisting of 7 unimodal functions (F1 to F7) and 16 multimodal functions (F8 to F23). A full description of these benchmark functions is available in [78].

Figure 3 illustrates the exploration–exploitation ratio of the BOA method throughout the iteration process, offering visual support for analyzing how the algorithm balances global and local search strategies. Also, the results of the analysis of population diversity, exploration, and exploitation are reported in Table 1. The simulation results show that the BOA has favorable population diversity, where it has high values in the first iteration, while the values of this index are low in the last iteration. Also, based on the obtained results, in most cases the exploration–exploitation ratio of the BOA is close to 0.00%:100%. The findings obtained from this analysis confirm that the proposed BOA approach, by creating the appropriate population diversity during the iterations of the algorithm, provides a favorable performance in managing exploration and exploitation, and in balancing them during the search process.

## 4. Simulation Studies and Results

In this section, the performance of the proposed BOA approach in handling optimization tasks is evaluated.

### 4.1. Performance Comparison

To assess the BOA’s effectiveness in addressing optimization problems, its results were juxtaposed with those of twelve prominent metaheuristic algorithms: the GA [31], PSO [11], GSA [40], TLBO [51], MVO [42], GWO [22], WOA [26], MPA [25], TSA [17], RSA [28], AVOA [23], and WSO [21]. These twelve algorithms were selected from the numerous algorithms available in the literature. The reasons for choosing these twelve algorithms were as follows: the GA and PSO are among the first and most famous metaheuristic algorithms. The GSA, TLBO, MVO, GWO, and WOA are among the most cited metaheuristic algorithms that have been used in various optimization applications. The MPA, TSA, RSA, AVOA, and WSO approaches are among the recently published successful metaheuristic algorithms that have attracted the attention of many researchers in this short period of time. Comparing the proposed BOA approach with these twelve selected metaheuristic algorithms is a valuable comparison, after which the efficiency of the BOA will have been tested well. Table 2 outlines the control parameter values for the competing algorithms. The evaluation of the simulation results incorporates six statistical metrics: mean, best, worst, standard deviation (std), median, and rank. The mean index values were utilized for ranking the metaheuristic algorithms concerning each benchmark function.

### 4.2. Evaluation of the CEC 2017 Test Suite

In this section, the performance of the BOA and competing algorithms is evaluated using the CEC 2017 test suite, considering problem dimensions (number of decision variables) equal to 10, 30, 50, and 100. The CEC 2017 test suite comprises thirty benchmark functions, including three unimodal functions (C17-F1 to C17-F3), seven multimodal functions (C17-F4 to C17-F10), ten hybrid functions (C17-F11 to C17-F20), and ten composition functions (C17-F21 to C17-F30). The C17-F2 function is excluded due to its unstable behavior, as described in [79].

The results of employing the BOA approach and competing algorithms on the CEC 2017 test suite are presented in Table 3. Boxplot diagrams depicting the performance of the BOA and competing algorithms in optimizing the CEC 2017 test suite are illustrated in Figure 4. The outcomes indicate that the BOA outperformed other optimizers, ranking as the top performer for functions C17-F1, C17-F3 to C17-F21, C17-F23, C17-F24, and C17-F27 to C17-F30.

Overall, the BOA demonstrated its efficacy in providing effective solutions for the CEC 2017 test suite, showcasing a commendable ability to explore, exploit, and maintain balance throughout the search process. The simulation results establish the BOA’s superior performance over competing algorithms, securing the top rank as the best optimizer for handling the CEC 2017 test suite.

### 4.3. Statistical Analysis

In this section, a statistical analysis was performed on the performances of the BOA and rival algorithms to assess the significance of the BOA’s superiority from a statistical perspective. The Wilcoxon signed-rank test [80], a non-parametric test for matched or paired data, was employed for this purpose. This test helps determine whether there is a significant difference between the averages of two data samples. The results of the Wilcoxon signed-rank test, presented in Table 4, indicate instances where the BOA exhibits statistically significant superiority over the respective competing algorithms, with a *p*-value criterion of less than 0.05.

### 4.4. Discussion

In this subsection, the performance of the BOA compared to competing algorithms is discussed. The CEC 2017 test suite has different types of objective functions.

Unimodal functions C17-F1 and C17-F3 have only one main optimum (i.e., global optimum), and for that reason they are suitable criteria for measuring the exploitation ability of metaheuristic algorithms. Analysis of the simulation results shows that the proposed BOA approach, with a strong performance in local search, has superior performance against all twelve competing algorithms for handling unimodal functions. Therefore, as the first strength, the superiority of the BOA in exploitation is confirmed against competing algorithms.

Multimodal functions C17-F4 to C17-F10, in addition to the main optimum (i.e., the global optimum), also have a number of local optima, which challenge the exploration ability of metaheuristic algorithms. The findings obtained from the simulation results show that the BOA, with global search management, was able to achieve the rank of the best optimizer in the competition with the compared algorithms to handle the functions C17-F4 to C17-F10. The simulation results confirm that, as the second strength, the BOA has a better exploration ability to manage global search compared to competing algorithms.

Hybrid functions C17-F11 to C17-F20 and composition functions C17-F21 to C17-F30 are complex optimization problems that challenge the performance of metaheuristic algorithms in establishing a balance between exploration and exploitation. The simulation results of these functions show that the BOA was able to achieve the rank of the best optimizer in most of these benchmark functions, except for C17-F22, C17-F25, and C17-F26. The simulation results confirm that the BOA is highly capable of balancing exploration and exploitation when facing complex optimization problems. Therefore, as a third strength, the superiority of the BOA in balancing exploration and exploitation is confirmed compared to competing algorithms.

In addition, the statistical analysis of the Wilcoxon signed-rank test and the values obtained for the p-value index, as the fourth strength, confirm that the BOA has a significant statistical superiority compared to all twelve competing algorithms.

## 5. BOA for Real-World Applications

In this section, the effectiveness of the proposed BOA approach in addressing real-world optimization tasks is evaluated. To this end, twenty-two constrained optimization problems from the CEC 2011 test suite, along with four engineering design problems, are utilized.

### 5.1. Evaluation of CEC 2011 Test Suite

In this subsection, the performance of the BOA in optimizing the CEC 2011 test suite, which comprises twenty-two constrained optimization problems from real-world applications, is assessed. Detailed descriptions and information about the CEC 2011 test suite can be found in [81]. The results of employing the BOA and competing algorithms on the CEC 2011 test suite are presented in Table 5, and the boxplot diagrams illustrating the performance of the BOA and competing algorithms are depicted in Figure 5. The optimization outcomes highlight that the BOA effectively generated suitable solutions for this test suite, showcasing a balanced exploration and exploitation throughout the search process. Notably, the BOA emerges as the top optimizer for solving functions C11-F1 to C11-F22, demonstrating superior performance in comparison to competing algorithms. Statistical analysis, specifically the Wilcoxon signed-rank test, further validates the significant statistical superiority of the BOA in these evaluations.

### 5.2. Pressure Vessel Design Problem

The design of the pressure vessel in engineering aims primarily to minimize construction costs, as illustrated in Figure 6. The mathematical representation of pressure vessel design is defined as follows [82]:

Consider: $X=\left[x_{1},x_{2},x_{3},x_{4}\right]=\left[T_{s},T_{h},R,L\right]$.

Minimize: $f\left(x\right)=0.6224x_{1}x_{3}x_{4}+1.778x_{2}x_{3}^{2}+3.1661x_{1}^{2}x_{4}+19.84x_{1}^{2}x_{3}.$

Subject to
$$g_{1}\left(x\right)=-x_{1}+0.0193x_{3}\;\leq \;0,\;g_{2}\left(x\right)=-x_{2}+0.00954x_{3}\leq \;0,$$
$$g_{3}\left(x\right)=-\pi x_{3}^{2}x_{4}-\frac{4}{3}\pi x_{3}^{3}+1296000\leq \;0,\;g_{4}\left(x\right)=x_{4}-240\;\leq \;0.$$
with
$$0\leq x_{1},x_{2}\leq 100\;{and\;10\leq x}_{3},x_{4}\leq 200.$$

The outcomes derived from applying the BOA and rival algorithms to optimize pressure vessel design are documented in Table 6 and Table 7. According to the results, the BOA yielded the optimal solution for this design, with design variable values of (0.7781685, 0.3846492, 40.319615, 200) and an objective function value of 5885.3263. The convergence curve of the BOA throughout the discovery of the optimal solution for pressure vessel design is depicted in Figure 7. Examination of the optimization results indicates that the BOA exhibits superior performance in addressing pressure vessel design challenges, outperforming competing algorithms.

### 5.3. Speed Reducer Design Problem

The design of a speed reducer is a practical engineering application focused on minimizing the weight of the speed reducer, as illustrated in Figure 8. The mathematical model for the design of the speed reducer is outlined in [83,84]:

Consider: $X=\left[x_{1,}x_{2},x_{3},x_{4},x_{5}{,x}_{6},x_{7}\right]=\left[b,m,p,l_{1},l_{2},d_{1},d_{2}\right]$.

Minimize: $f\left(x\right)=0.7854x_{1}x_{2}^{2}\left(3.3333x_{3}^{2}+14.9334x_{3}-43.0934\right)-1.508x_{1}\left(x_{6}^{2}+x_{7}^{2}\right)+7.4777\left(x_{6}^{3}+x_{7}^{3}\right)+0.7854(x_{4}x_{6}^{2}+x_{5}x_{7}^{2})$.

Subject to
$$g_{1}\left(x\right)=\frac{27}{x_{1}x_{2}^{2}x_{3}}-1\leq 0,g_{2}\left(x\right)=\frac{397.5}{x_{1}x_{2}^{2}x_{3}^{2}}-1\leq 0,$$
$$g_{3}\left(x\right)=\frac{1.93x_{4}^{3}}{x_{2}x_{3}x_{6}^{4}}-1\leq 0,g_{4}\left(x\right)=\frac{1.93x_{5}^{3}}{x_{2}x_{3}x_{7}^{4}}-1\leq 0,$$
$$g_{5}\left(x\right)=\frac{1}{110x_{6}^{3}}\sqrt{{\left(\frac{745x_{4}}{x_{2}x_{3}}\right)}^{2}+16.9\times 10^{6}}-1\leq 0,$$
$$g_{6}(x)=\frac{1}{85x_{7}^{3}}\sqrt{{\left(\frac{745x_{5}}{x_{2}x_{3}}\right)}^{2}+157.5\times 10^{6}}-1\leq 0,$$
$$g_{7}\left(x\right)=\frac{x_{2}x_{3}}{40}-1\leq 0,g_{8}\left(x\right)=\frac{{5x}_{2}}{x_{1}}-1\leq 0,$$
$$g_{9}\left(x\right)=\frac{x_{1}}{12x_{2}}-1\leq 0,g_{10}\left(x\right)=\frac{{1.5x}_{6}+1.9}{x_{4}}-1\leq 0,$$
$$g_{11}\left(x\right)=\frac{{1.1x}_{7}+1.9}{x_{5}}-1\leq 0.$$
with
$$2.6\leq x_{1}\leq 3.6,0.7\leq x_{2}\leq 0.8,17\leq x_{3}\leq 28,7.3\leq x_{4}\leq 8.3,7.8\leq x_{5}\leq 8.3,2.9\leq x_{6}\leq 3.9,\;and\;5\leq x_{7}\leq 5.5.$$

The outcomes of implementing the BOA and competing optimizers to address the speed reducer design challenges are documented in Table 8 and Table 9. The BOA yielded the optimal solution for this design, characterized by design variable values (3.5, 0.7, 17, 7.3, 7.8, 3.3502147, 5.2866832) and an objective function value of 2996.3482. The convergence curve, depicting the BOA’s performance in optimizing the speed reducer design, is illustrated in Figure 9. The analysis of the simulation results confirms that the BOA demonstrated more effective performance in tackling the speed reducer design compared to its competitors.

### 5.4. Welded Beam Design

The design of a welded beam poses a real-world engineering challenge, intending to minimize the fabrication cost of the beam, as depicted in Figure 10. The mathematical model governing the welded beam design is outlined as follows [26]:

Consider: $X=\left[x_{1},x_{2},x_{3},x_{4}\right]=\left[h,l,t,b\right]$.

Minimize: $f(x)=1.10471x_{1}^{2}x_{2}+0.04811x_{3}x_{4}(14.0+x_{2})$.

Subject to
$$g_{1}\left(x\right)=\tau \left(x\right)-13600\leq 0,\;g_{2}\left(x\right)=\sigma \left(x\right)-30000\;\leq \;0,$$
$$g_{3}\left(x\right)=x_{1}-x_{4}\leq \;0,\;g_{4}(x)=0.10471x_{1}^{2}+0.04811x_{3}x_{4}\;(14+x_{2})-5.0\;\leq \;0,$$
$$g_{5}\left(x\right)=0.125\;-x_{1}\leq \;0,\;g_{6}\left(x\right)=\delta \;\left(x\right)-0.25\;\leq \;0,$$
$$g_{7}\left(x\right)=6000-p_{c}\;\left(x\right)\leq \;0,$$
where
$$\tau \left(x\right)=\sqrt{{\left(\tau^{\prime }\right)}^{2}+\left(2\tau \tau^{\prime }\right)\frac{x_{2}}{2R}+{\left(\tau \prime \prime \right)}^{2}\;},\;\tau^{\prime }=\frac{6000}{\sqrt{2}x_{1}x_{2}},\;\tau^{\prime \prime }=\frac{MR}{J},$$
$$M=6000\left(14+\frac{x_{2}}{2}\right),\;R=\sqrt{\frac{x_{2}^{2}}{4}+{\left(\frac{x_{1}+x_{3}}{2}\right)}^{2}},$$
$$J=2x_{1}x_{2}\sqrt{2}\left(\frac{x_{2}^{2}}{12}+{\left(\frac{x_{1}+x_{3}}{2}\right)}^{2}\right),\;\sigma \left(x\right)=\frac{504000}{x_{4}x_{3}^{2}},\;\delta \left(x\right)=\frac{2.1925}{x_{4}x_{3}^{3}},$$
$$p_{c}\;\left(x\right)=17062.0748\cdot x_{3}x_{4}^{3}\left(1-\frac{x_{3}}{28}\sqrt{\frac{5}{8}}\right).$$
with $0.1\leq x_{1},x_{4}\leq 2\;and\;0.1\leq x_{2},x_{3}\leq 10.$

The optimization outcomes for the welded beam design, utilizing the BOA and competing algorithms, are outlined in Table 10 and Table 11. The BOA yielded the optimal solution for this design, with design variable values set at (0.2057296, 3.4704887, 9.0366239, 0.2057296), resulting in an objective function value of 1.7246798. The convergence process of the BOA towards the optimal solution for the welded beam design is illustrated in Figure 11. The simulation results underscore the effectiveness of the BOA in addressing the welded beam design problem, showcasing superior performance compared to competing algorithms.

### 5.5. Tension/Compression Spring Design Problem

The engineering challenge in tension/compression spring design is to minimize the weight of the spring, as depicted in Figure 12. The mathematical model for tension/compression spring design is outlined as follows [26]:

*Consider:* $X=\left[x_{1},x_{2},x_{3}\right]=\left[d,D,P\right].$

*Minimize*: $f\left(x\right)=\left(x_{3}+2\right)x_{2}x_{1}^{2}.$

*Subject to*$$g_{1}\left(x\right)=1-\frac{x_{2}^{3}x_{3}}{71785x_{1}^{4}}\;\leq \;0,\;g_{2}\left(x\right)=\frac{4x_{2}^{2}-x_{1}x_{2}}{12566(x_{2}x_{1}^{3})}+\frac{1}{5108x_{1}^{2}}-1\leq \;0,$$$$g_{3}\left(x\right)=1-\frac{140.45x_{1}}{x_{2}^{2}x_{3}}\leq \;0,\;g_{4}\left(x\right)=\frac{x_{1}+x_{2}}{1.5}-1\;\leq \;0.$$
with $0.05\leq x_{1}\leq 2,{0.25\leq x}_{2}\leq 1.3\;and\;2\leq x_{3}\leq 15$.

The optimization outcomes for tension/compression spring design using the BOA and competing algorithms are outlined in Table 12 and Table 13. The BOA yielded the optimal solution for this design, with design variable values of (0.0516891, 0.3567177, 11.288966) and an objective function value of 0.0126019. The convergence curve depicting the BOA’s performance in optimizing the tension/compression spring design is illustrated in Figure 13. The simulation results demonstrate that the BOA exhibited superior performance compared to competing algorithms by delivering improved outcomes for tension/compression spring design.

## 6. Conclusions and Future Works

In this paper, motivated by the No Free Lunch (NFL) theorem, a new human-based metaheuristic algorithm called the Botox Optimization Algorithm (BOA) was introduced, mimicking the human action of Botox injections. The originality of the proposed BOA approach was confirmed based on the best knowledge obtained from the literature review, where no metaheuristic algorithm based on Botox injection modeling has been designed so far. The fundamental inspiration of the BOA is the injection of Botox into the areas of the face in order to remove defects from the face and increase facial beauty. The theory of the BOA was stated, and the various stages of its implementation were mathematically modeled based on the simulation of Botox injection. The performance of the BOA was evaluated on the CEC 2017 test suite. The optimization results showed that the BOA has a high ability to balance exploration and exploitation during the search process. To measure the quality of the BOA, the obtained results were compared with the performance of twelve well-known metaheuristic algorithms. The simulation results showed that the BOA outperformed competing algorithms by providing better results in most benchmark functions. Using statistical analysis, it was shown that the BOA has significant statistical superiority over competing algorithms. Also, the implementation of the BOA on twenty-two constrained optimization problems from the CEC 2011 test suite showed the ability of the proposed approach to handle real-world applications.

After introducing the proposed BOA approach, several research paths can be considered for further studies:Binary BOA: The real version of the BOA is detailed and explained thoroughly in this paper. Nonetheless, many scientific optimization issues, like feature selection, require the use of binary versions of metaheuristic algorithms for efficient optimization. Consequently, developing the binary version of the BOA (BBOA) is a notable focus of this research.Multi-objective BOA: Optimization problems are classified based on the number of objective functions, which are either single-objective or multi-objective. To find an optimal solution, many problems require the consideration of multiple objective functions simultaneously. Hence, exploring the potential of developing a multi-objective version of the BOA (MOBOA) to address multi-objective optimization dilemmas is another area of research highlighted in this paper.Hybrid BOA: Researchers have always been intrigued by the idea of merging multiple metaheuristic algorithms to leverage the strengths of each and establish a more efficient hybrid strategy. Hence, a potential future research endeavor includes crafting hybrid versions of the BOA.Tackle new domains: Exploring opportunities for employing the BOA in tackling practical applications and optimizing problems within various scientific fields, like robotics, renewable energy, chemical engineering, and image processing, is a focus for future research proposals.

## Figures and Tables

**Figure 1 biomimetics-09-00137-f001:**
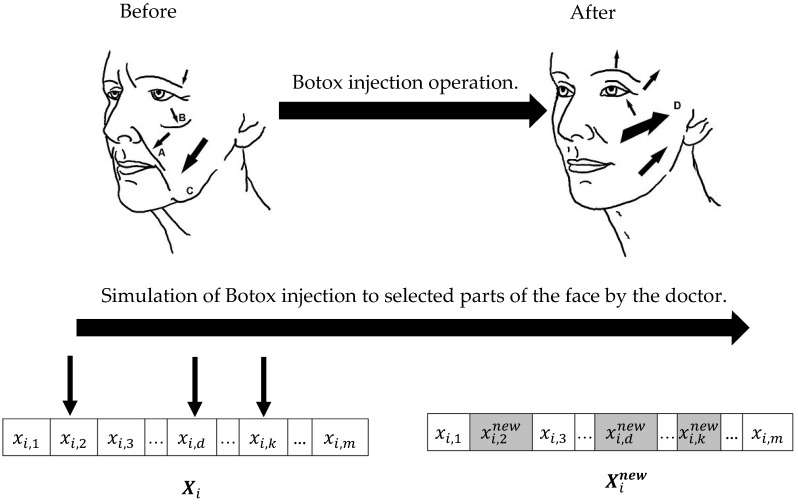
Schematic diagram of the Botox injection and the proposed BOA.

**Figure 2 biomimetics-09-00137-f002:**
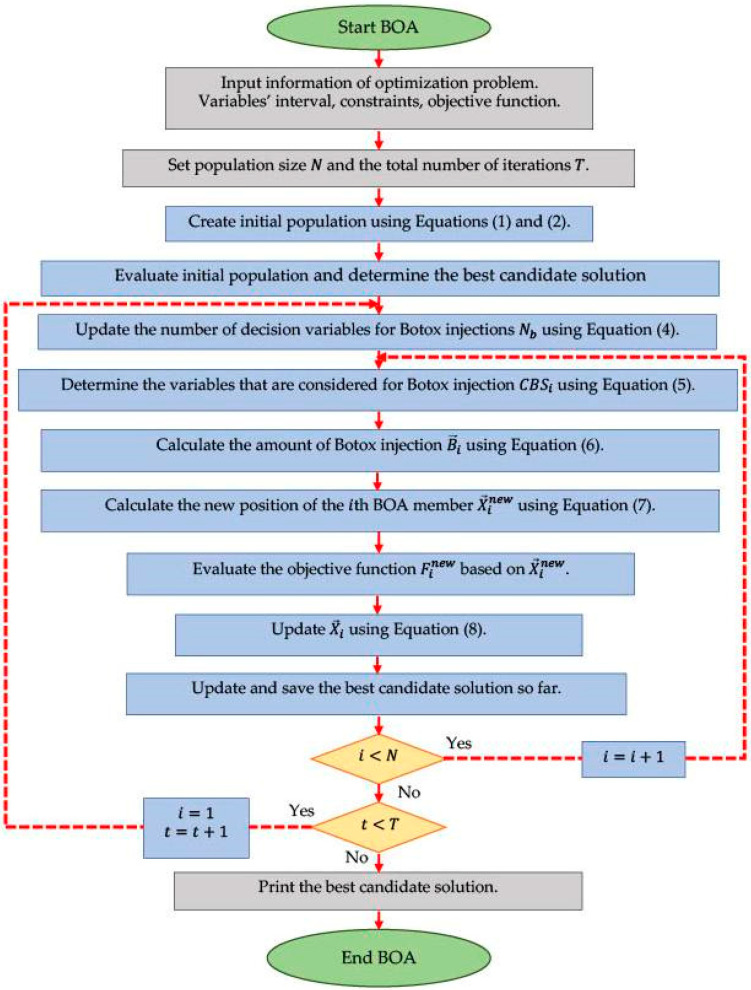
Flowchart of the BOA.

**Figure 3 biomimetics-09-00137-f003:**
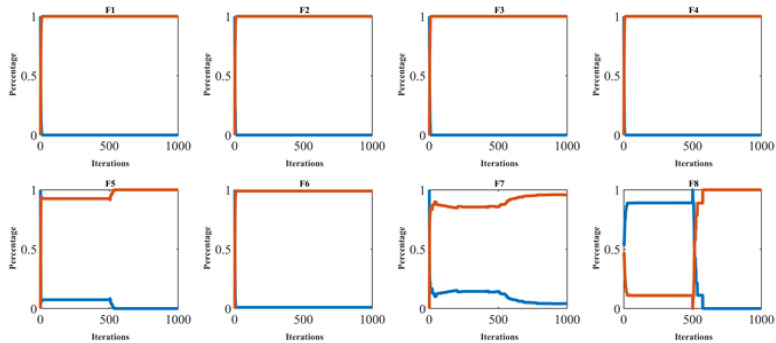
Exploration and exploitation of the BOA.

**Figure 4 biomimetics-09-00137-f004:**
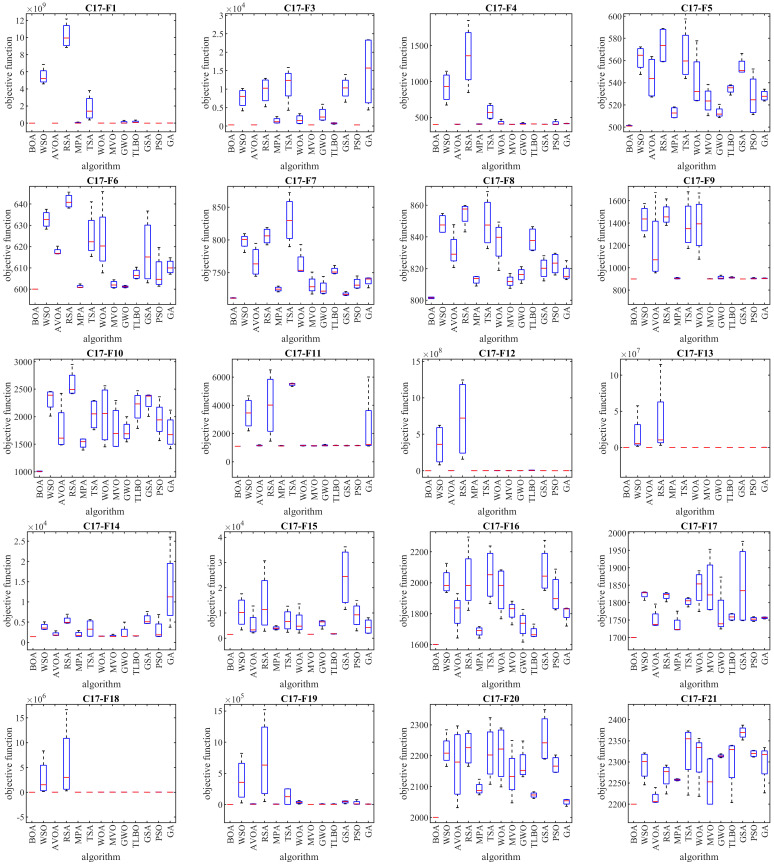
Boxplot representations illustrating the performances of the BOA and rival algorithms on the CEC 2017 test suite.

**Figure 5 biomimetics-09-00137-f005:**
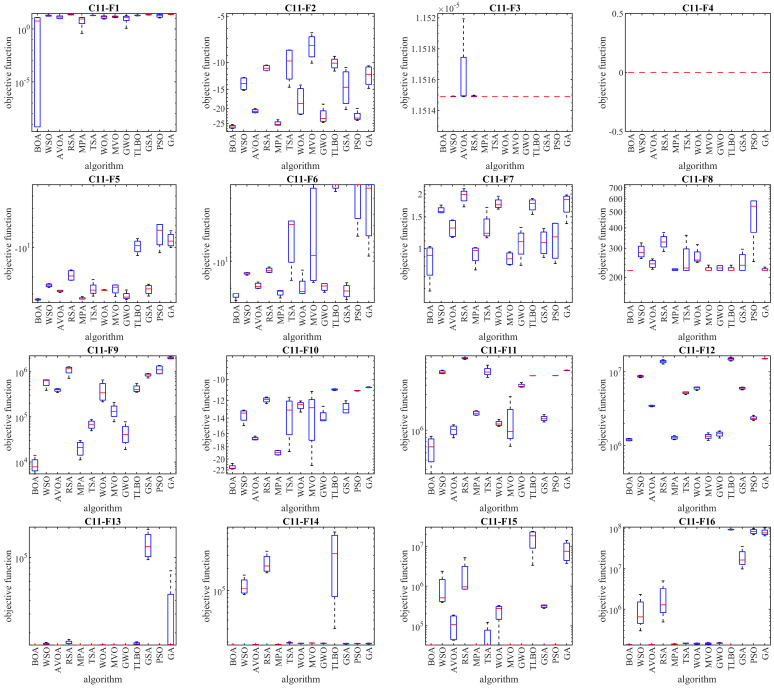
Boxplot diagrams of the BOA and competing algorithms’ performances on the CEC 2011 test suite.

**Figure 6 biomimetics-09-00137-f006:**
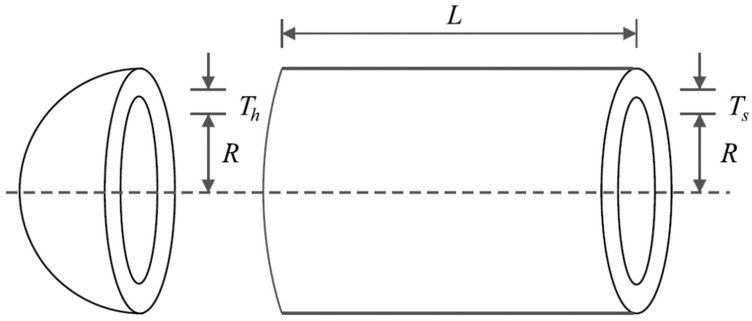
Schematic of pressure vessel design. The thickness of the shell is $T_{s}$, the thickness of the head is $T_{h}$, the length of cylindrical shell is L, and the inner radius is R.

**Figure 7 biomimetics-09-00137-f007:**
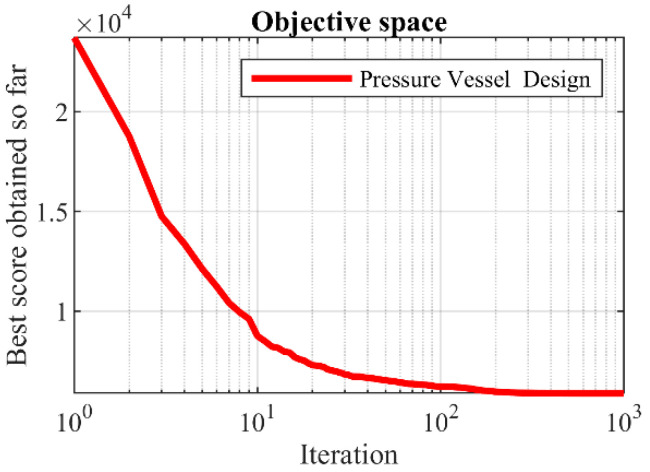
The BOA’s performance convergence curve on pressure vessel design.

**Figure 8 biomimetics-09-00137-f008:**
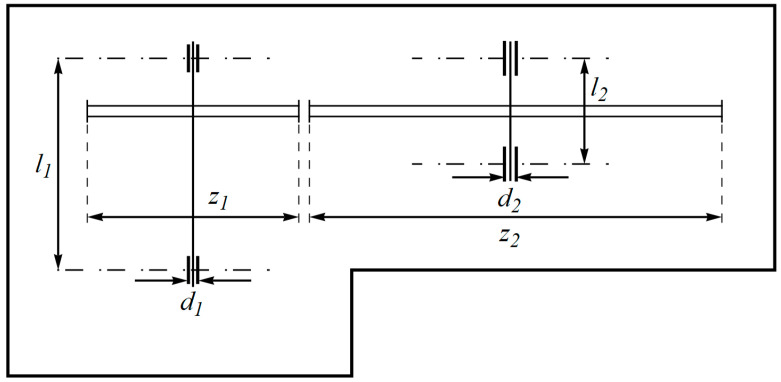
Schematic of speed reducer design. The face width is b, the number of teeth on the pinion is z, the module of teeth is m, the length of the second shaft between bearings is $l_{2}$, the length of the first shaft between bearings is $l_{1}$, the second shaft’s diameter is $d_{2}$, and the first shaft’s diameter is $d_{1}$.

**Figure 9 biomimetics-09-00137-f009:**
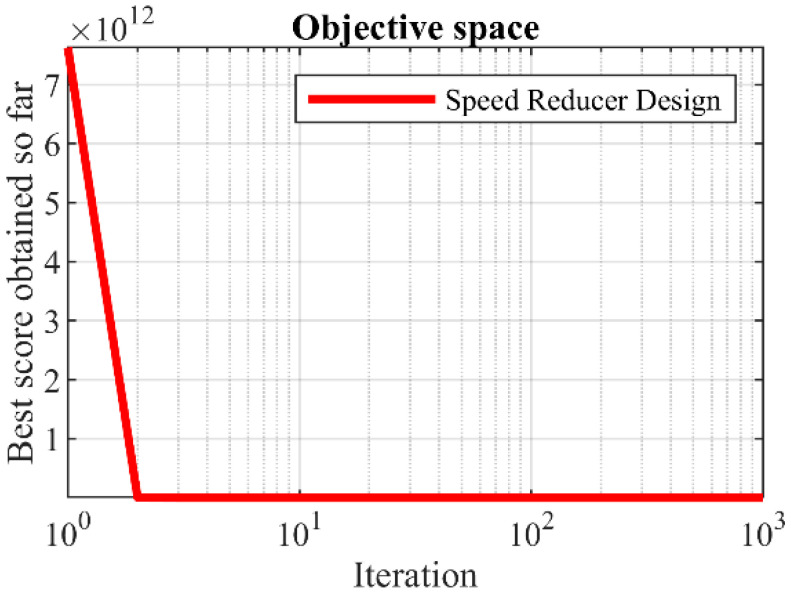
The BOA’s performance convergence curve on speed reducer design.

**Figure 10 biomimetics-09-00137-f010:**
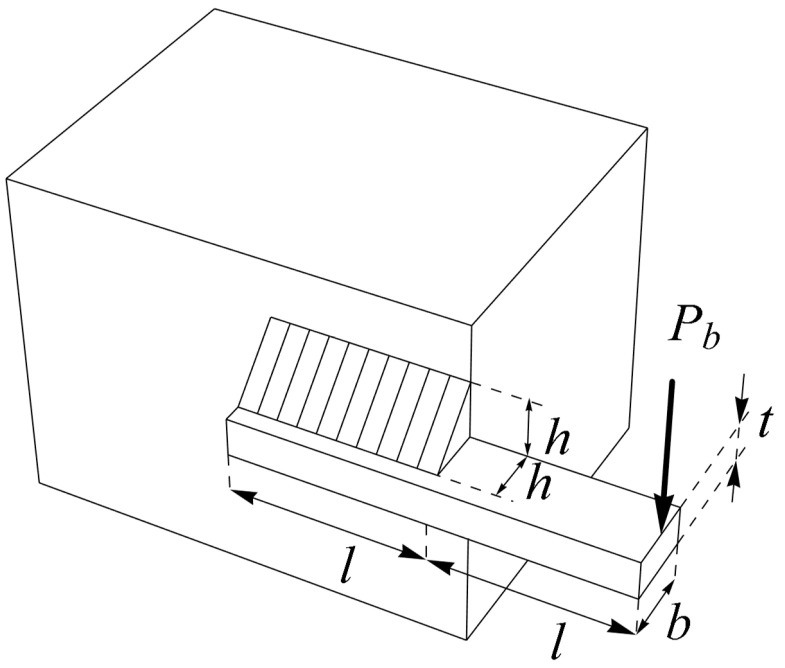
Schematic of welded beam design. The bar height is t, the weld thickness is h, the thickness of bar is b, and the length of clamped bar is l.

**Figure 11 biomimetics-09-00137-f011:**
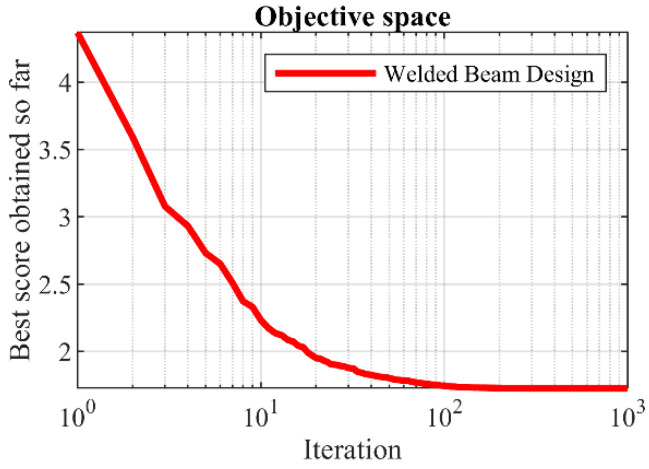
The BOA’s performance convergence curve on welded beam design.

**Figure 12 biomimetics-09-00137-f012:**
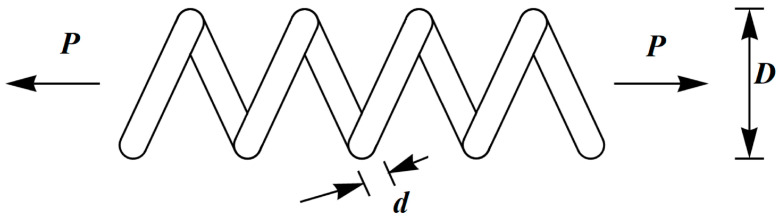
Schematic of tension/compression spring design. The wire’s diameter is d, the number of active coils is P, and the mean coil’s diameter is D.

**Figure 13 biomimetics-09-00137-f013:**
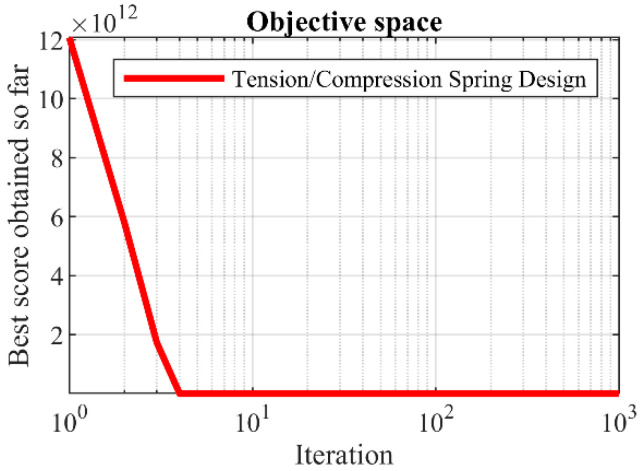
The BOA’s performance convergence curve on tension/compression spring design.

**Table 1 biomimetics-09-00137-t001:** Population diversity, exploration, and exploitation percentage results.

Function Name	Exploration	Exploitation	Diversity	
			First Iteration	Last Iteration
F1	0	1	140.5903	0
F2	0	1	10.94242	0
F3	0	1	252.3905	0
F4	0	1	128.8175	0
F5	0	1	44.92006	0
F6	0.012744	0.987256	114.0171	1.453073
F7	0.049372	0.950628	1.518403	0.074966
F8	5.84E-10	1	1230.61	1.19E-06
F9	4.76E-10	1	9.555822	4.55E-09
F10	1.8E-17	1	50.32814	9.04E-16
F11	3.7E-11	1	885.0467	3.27E-08
F12	0	1	61.45876	0
F13	0	1	77.58755	0
F14	2.02E-08	1	23.65722	4.77E-07
F15	5.93E-11	1	4.048837	2.4E-10
F16	0.082221	0.917779	1.61018	0.13239
F17	6.68E-10	1	4.961695	3.31E-09
F18	0.068118	0.931882	0.757968	0.051631
F19	0.245007	0.754993	0.378584	0.118559
F20	0.054777	0.945223	0.441119	0.024163
F21	1.95E-10	1	3.149125	7.25E-10
F22	1.36E-10	1	3.505294	6.63E-10
F23	9.32E-11	1	4.347473	4.27E-10

**Table 2 biomimetics-09-00137-t002:** Control parameters’ values.

Algorithm	Parameter	Value
GA		
	Type	Real coded
	Selection	Roulette wheel (proportionate)
	Crossover	Whole arithmetic (Probability=0.8, $\alpha \in \left[-0.5,1.5\right]$)
	Mutation	Gaussian (Probability=0.05)
PSO		
	Topology	Fully connected
	Cognitive and social constant	($C_{1},\;C_{2}$) =(2,2)
	Inertia weight	Linear reduction from 0.9 to 0.1
	Velocity limit	10% of dimension range
GSA		
	Alpha, $G_{0}$, $R_{norm}$, $R_{power}$	20, 100, 2, 1
TLBO		
	$T_{F}$: Teaching factor	$T_{F}=round\;\;\left[\left(1+rand\right)\right]$
	Random number	*rand* is a random number from $\left[0,1\right].$
GWO		
	Convergence parameter (a)	a: Linear reduction from 2 to 0.
MVO		
	Wormhole Existence Probability (WEP)	Min(WEP)=0.2 and Max(WEP)=1.
	Exploitation accuracy over the iterations (p)	p=6.
WOA		
	Convergence parameter (a)	a: Linear reduction from 2 to 0.
	*r* is a random vector in $\left[0,\;1\right]$;	
	*l* is a random number in $\left[-1,\;1\right]$	
TSA		
	$P_{min},P_{max}$	1, 4
	$c_{1},c_{2},\;c_{3}$	Random numbers lie in the interval $\left[0,1\right].$
MPA		
	Constant number	P=0.5
	Random vector	*R* is a vector of uniform random numbers in $\left[0,1\right].$
	Fish Aggregating Devices (*FADs*)	FADs=0.2
	Binary vector	U=0 or 1
RSA		
	Sensitive parameter	β=0.01
	Sensitive parameter	α=0.1
	Evolutionary Sense (ES)	ES: Randomly decreasing values between 2 and −2
AVOA		
	$L_{1},L_{2}$	0.8, 0.2
	w	2.5
	$P_{1},P_{2},P_{3}$	0.6, 0.4, 0.6
WSO		
	$F_{min},F_{max}$	0.07, 0.75
	τ, $a_{0},a_{1},a_{2}$	4.125, 6.25, 100, 0.0005

**Table 3 biomimetics-09-00137-t003:** Optimization results of the CEC 2017 test suite.

		BOA	WSO	AVOA	RSA	MPA	TSA	WOA	MVO	GWO	TLBO	GSA	PSO	GA
C17-F1	Mean	**100**	5.46E+09	3848.625	1.02E+10	35,331,826	1.74E+09	6,458,531	7530.831	88,328,653	1.47E+08	747.4345	3148.604	11,867,816
	Best	**100**	4.59E+09	115.6391	8.84E+09	11,218.07	3.73E+08	4,702,752	4790.099	27,833.68	65,653,192	100.0193	345.9935	6,145,607
	Worst	**100**	6.85E+09	11,928.77	1.22E+10	1.28E+08	3.8E+09	8,503,451	11,096.78	3.21E+08	3.56E+08	1792.381	9323.402	17,037,274
	Std	**0**	1.07E+09	6005.345	1.64E+09	67,796,179	1.66E+09	1,750,529	3215.353	1.69E+08	1.52E+08	796.8781	4524.035	4,954,471
	Median	**100**	5.21E+09	1675.046	9.94E+09	6,476,105	1.4E+09	6,313,960	7118.225	16,188,754	84,181,901	548.6691	1462.51	12,144,191
	Rank	**1**	12	4	13	8	11	6	5	9	10	2	3	7
C17-F3	Mean	**300**	7591.42	301.8957	9658.226	1408.746	11,214.42	1731.412	300.0547	3071.866	726.7343	10,268.87	**300**	14,789.2
	Best	**300**	4113.783	**300**	5207.518	791.8459	4270.307	619.6359	300.0127	1529.615	471.4213	6461.812	**300**	4354.021
	Worst	**300**	10,174.7	304.0548	12,922.2	2537.378	15,855.2	3333.866	300.1245	5893.445	893.5147	13,957.47	**300**	23,376.32
	Std	**0**	2896.121	2.393662	3838.968	876.4573	5353.91	1391.667	0.05345	2191.143	201.3742	3364.571	5.05E-14	10,814.33
	Median	**300**	8038.595	301.764	10,251.59	1152.881	12,366.08	1486.072	300.0407	2432.202	771.0005	10,328.09	**300**	15,713.23
	Rank	**1**	9	4	10	6	12	7	3	8	5	11	2	13
C17-F4	Mean	**400**	919.2957	404.7605	1352.77	406.7394	576.7582	425.1975	403.341	411.7605	409.1884	404.5619	420.3519	414.7475
	Best	**400**	672.5461	401.2436	845.7611	402.4511	477.9916	406.4543	401.5971	406.1014	408.4021	403.5684	400.1059	411.7011
	Worst	**400**	1142.743	406.5393	1849.389	411.4014	692.0754	473.6998	404.9048	428.4155	409.6849	406.0879	470.5109	418.4747
	Std	**0**	229.2926	2.715361	466.2296	4.804772	114.2038	35.29756	1.871726	12.08414	0.598616	1.257481	36.76729	3.226255
	Median	**400**	930.9465	405.6296	1357.964	406.5525	568.4828	410.3179	403.431	406.2626	409.3334	404.2956	405.3954	414.407
	Rank	**1**	12	4	13	5	11	10	2	7	6	3	9	8
C17-F5	Mean	**501.2464**	562.2888	544.5598	573.6638	513.037	565.1128	541.4479	523.9769	513.1801	534.4525	554.4872	528.2287	528.3418
	Best	**500.9951**	547.4308	527.1499	558.8475	508.4371	543.7341	523.7238	510.3406	508.6157	528.9014	549.5681	511.27	523.5807
	Worst	**501.9917**	572.2487	563.5795	588.8477	518.2122	597.5197	577.7367	538.4222	520.5555	538.0275	566.3814	552.3704	534.175
	Std	**0.540776**	12.11165	20.81022	18.12589	5.591417	25.96825	27.5346	12.761	5.604703	4.364766	8.746105	20.64316	5.206467
	Median	**500.9993**	564.7379	543.7549	573.48	512.7494	559.5987	532.1655	523.5724	511.7746	535.4404	550.9996	524.6372	527.8058
	Rank	**1**	11	9	13	2	12	8	4	3	7	10	5	6
C17-F6	Mean	**600**	632.7824	617.595	641.3535	601.2128	625.225	623.5356	602.184	601.1449	606.9718	617.4791	607.548	610.4234
	Best	**600**	628.194	616.5747	638.0899	600.7222	615.3143	607.646	600.4796	600.6055	604.8344	602.9627	601.3762	607.0149
	Worst	**600**	637.5379	620.1871	645.6746	602.4363	641.0634	645.9187	604.3818	601.7463	610.3045	636.7176	619.5657	614.7356
	Std	**0**	4.413003	1.88613	3.709445	0.889975	12.08883	17.5391	1.908475	0.513952	2.714097	16.99487	8.979175	3.72455
	Median	**600**	632.699	616.8091	640.8248	600.8464	622.2611	620.2889	601.9373	601.1139	606.3741	615.1181	604.6251	609.9715
	Rank	**1**	12	9	13	3	11	10	4	2	5	8	6	7
C17-F7	Mean	**711.1267**	797.8688	766.4303	805.8543	724.8556	830.358	762.9002	731.196	726.2532	752.71	717.2159	733.0903	737.2869
	Best	**710.6726**	780.6589	744.4304	792.4154	720.5783	789.6269	751.7383	717.2818	717.5639	748.1129	714.8806	725.8012	726.7599
	Worst	**711.7995**	809.4588	794.6393	818.7295	729.351	872.5061	792.8006	750.7791	744.0606	760.9719	721.0193	744.8461	741.9507
	Std	**0.557384**	13.284	25.13695	13.44422	4.022787	39.18582	21.75715	15.33828	13.2473	6.2627	2.892038	9.451487	7.758608
	Median	**711.0174**	800.6788	763.3257	806.1362	724.7466	829.6494	753.531	728.3615	721.6942	750.8777	716.4818	730.8569	740.2185
	Rank	**1**	11	10	12	3	13	9	5	4	8	2	6	7
C17-F8	Mean	**801.4928**	848.1769	831.617	854.5622	812.8571	849.0636	836.9437	812.0059	816.0909	838.313	820.1744	823.1267	817.0493
	Best	**800.995**	842.8959	820.611	843.1502	808.9812	832.594	818.8766	807.536	810.6855	831.2975	812.2041	815.9405	813.0058
	Worst	**801.9912**	854.7381	847.6997	859.9005	815.0312	868.6915	849.3401	816.8821	821.1703	846.4472	828.0838	829.679	824.9831
	Std	**0.625636**	6.701009	12.45098	8.409374	3.041009	17.47606	14.23885	4.180792	4.773722	8.432783	7.354204	7.407172	5.86208
	Median	**801.4926**	847.5368	829.0787	857.599	813.708	847.4845	839.779	811.8029	816.2538	837.7536	820.2048	823.4438	815.1042
	Rank	**1**	11	8	13	3	12	9	2	4	10	6	7	5
C17-F9	Mean	**900**	1430.961	1192.849	1476.617	905.2837	1388.608	1383.102	900.8146	912.1313	912.0221	**900**	904.3122	905.1958
	Best	**900**	1276.126	954.6109	1379.243	900.3329	1172.43	1076.987	900.0011	900.5827	907.3517	**900**	900.9142	902.8443
	Worst	**900**	1573.852	1673.688	1616.361	913.5628	1681.601	1668.813	903.166	933.6785	920.3352	**900**	912.5228	909.2282
	Std	**0**	140.1941	362.6238	109.8787	6.480782	239.779	271.1443	1.706569	16.89776	6.209908	**0**	6.03293	3.14228
	Median	**900**	1436.932	1071.549	1455.432	903.6196	1350.2	1393.304	900.0457	907.1319	910.2007	**900**	901.9059	904.3554
	Rank	**1**	11	8	12	5	10	9	2	7	6	**1**	3	4
C17-F10	Mean	**1006.179**	2311.834	1782.706	2588.899	1519.782	2039.361	2031.708	1785.695	1729.938	2179.736	2286.565	1951.968	1720.13
	Best	**1000.284**	2010.815	1486.512	2416.757	1393.592	1762.237	1452.075	1458.762	1542.197	1786.243	2004.951	1563.92	1417.123
	Worst	**1012.668**	2456.623	2423.148	2951.232	1595.402	2293.271	2559.74	2290.911	1998.556	2469.964	2393.264	2360.873	2118.219
	Std	**7.244311**	225.7849	478.7803	270.7973	103.4757	304.9995	582.7437	438.7957	211.0887	316.3582	204.7244	356.2546	327.1839
	Median	**1005.882**	2389.949	1610.583	2493.804	1545.066	2050.967	2057.508	1696.554	1689.498	2231.369	2374.023	1941.539	1672.588
	Rank	**1**	12	5	13	2	9	8	6	4	10	11	7	3
C17-F11	Mean	**1100**	3442.37	1148.775	4000.383	1127.198	5484.031	1151.243	1127.661	1155.582	1151.197	1139.413	1143.772	2389.968
	Best	**1100**	2188.762	1117.145	1460.621	1113.275	5334.971	1113.032	1105.577	1121.743	1138.044	1119.755	1132.432	1115.129
	Worst	**1100**	4665.813	1202.357	6508.91	1159.111	5565.816	1173.513	1149.188	1229.094	1172.713	1169	1165.388	6006.61
	Std	**0**	1210.189	40.81842	2469.216	23.5556	111.6604	30.39483	23.71355	54.46268	16.28689	22.87379	16.15173	2624.636
	Median	**1100**	3457.454	1137.8	4016	1118.203	5517.668	1159.213	1127.94	1135.746	1147.016	1134.448	1138.634	1219.067
	Rank	**1**	11	6	12	2	13	8	3	9	7	4	5	10
C17-F12	Mean	**1352.959**	3.57E+08	1,109,704	7.11E+08	572,257.8	1,048,288	2,373,547	1,037,633	1,427,055	5,094,522	1,028,834	8145.64	610,018.9
	Best	**1318.646**	80,192,749	358,948.9	1.58E+08	20,015.65	543,606.5	173,159.8	8892.711	45,801.15	1,363,349	478,453.6	2528.033	176,684.6
	Worst	**1438.176**	6.23E+08	2,012,446	1.24E+09	895,571.3	1,286,990	3,937,641	3,259,364	2,233,764	9,018,848	1,739,931	14,021.19	1,076,854
	Std	**62.35801**	2.98E+08	841,688.2	5.98E+08	419,783.3	381,500.6	1,904,475	1,634,093	1,049,512	4,413,129	581,137.5	5698.792	402,260.9
	Median	**1327.506**	3.62E+08	1,033,710	7.22E+08	686,722.1	1,181,277	2,691,694	441,138.1	1,714,326	4,997,944	948,475.7	8016.671	593,268.4
	Rank	**1**	12	8	13	3	7	10	6	9	11	5	2	4
C17-F13	Mean	**1305.324**	17,335,628	18,471.94	34,661,545	5468.243	12,831.35	7630.328	6772.94	10,372.78	16,852.54	10,143.18	6664.754	54,887.32
	Best	**1303.114**	1,445,254	2734.676	2,877,682	3740.627	7639.26	3297.206	1386.726	6550.028	15,912.77	5078.007	2387.751	8603.051
	Worst	**1308.508**	57,542,238	31,658.65	1.15E+08	6689.907	20,335.57	15,267.01	12,470.28	14,494.81	19,148.28	14,291.03	16,841.17	181,516.7
	Std	**2.473462**	29,234,423	16,273.34	58,465,686	1530.879	5963.582	5938.403	6247.984	3543.419	1681.501	4237.976	7465.241	91,927.81
	Median	**1304.837**	5,177,510	19,747.21	10,349,214	5721.218	11,675.29	5978.549	6617.376	10,223.15	16,174.55	10,601.85	3715.048	14,714.78
	Rank	**1**	12	10	13	2	8	5	4	7	9	6	3	11
C17-F14	Mean	**1400.746**	3828.915	2027.645	5383.694	1945.167	3405.109	1520.154	1573.519	2355.252	1592.631	5604.306	3010.581	13,069.42
	Best	**1400**	3170.6	1681.663	4711.086	1435.363	1488.751	1482.596	1423.322	1462.902	1517.224	4631.803	1432.831	3748.476
	Worst	**1400.995**	5066.928	2841.448	6948.68	2917.489	5622.496	1560.295	1999.133	4996.706	1623.198	7611.074	6894.737	26,056.87
	Std	**0.541408**	945.4187	594.8446	1143.911	756.5397	2393.086	43.20222	308.8712	1916.5	54.97335	1519.18	2840.855	10,284.68
	Median	**1400.995**	3539.066	1793.735	4937.505	1713.908	3254.594	1518.862	1435.81	1480.701	1615.05	5087.174	1857.377	11,236.18
	Rank	**1**	10	6	11	5	9	2	3	7	4	12	8	13
C17-F15	Mean	**1500.331**	10,283.95	5333.31	13,989.15	3998.841	7053.662	6262.146	1542.234	5854.355	1711.148	24,088.27	9066.744	4578.338
	Best	**1500.001**	3259.43	2077.801	2745.559	3239.281	2327.191	2019.265	1526.157	3589.383	1584.901	11,316.55	2884.332	1894.197
	Worst	**1500.5**	17,559.48	12,730.21	30,627	4923.526	12,648.29	13,558.05	1554.385	6950.016	1801.652	36,159.61	14,917.69	8074.57
	Std	**0.256213**	6701.944	5408.362	13,248.87	760.3969	4827.04	5473.801	13.4196	1680.534	115.7645	12,916.31	5473.312	3344.03
	Median	**1500.413**	10,158.45	3262.612	11,292.03	3916.279	6619.585	4735.634	1544.197	6439.01	1729.02	24,438.46	9232.477	4172.293
	Rank	**1**	11	6	12	4	9	8	2	7	3	13	10	5
C17-F16	Mean	**1600.76**	2006.472	1811.471	2020.201	1684.989	2051.36	1953.664	1818.204	1729.658	1677.592	2077.466	1926.579	1804.186
	Best	**1600.356**	1936.677	1642.633	1821.325	1642.119	1864.782	1766.603	1727.666	1615.96	1651.389	1950.251	1824.477	1719.79
	Worst	**1601.12**	2125.766	1929.19	2296.53	1715.87	2237.602	2083.047	1880.447	1827.504	1732.365	2274.078	2087.786	1835.532
	Std	**0.343807**	91.7687	131.3759	218.4156	34.5268	184.0946	163.6931	70.32367	94.99226	41.07563	160.2418	132.7396	61.28572
	Median	**1600.781**	1981.722	1837.031	1981.476	1690.984	2051.528	1982.503	1832.352	1737.585	1663.306	2042.768	1897.026	1830.712
	Rank	**1**	10	6	11	3	12	9	7	4	2	13	8	5
C17-F17	Mean	**1700.099**	1823.352	1751.466	1819.495	1736.079	1803.161	1843.24	1844.13	1769.181	1758.941	1848.159	1752.864	1756.518
	Best	**1700.02**	1806.066	1734.73	1802.424	1722.121	1787.826	1774.309	1779.25	1724.691	1748.708	1748.396	1746.15	1753.36
	Worst	**1700.332**	1830.672	1795.908	1828.806	1775.629	1814.158	1891.142	1952.846	1873.227	1768.986	1975.689	1759.617	1758.979
	Std	**0.168864**	12.61303	32.32876	12.76453	28.70898	12.30981	55.23119	89.45313	75.87461	10.92649	126.1415	6.267908	2.763929
	Median	**1700.022**	1828.335	1737.613	1823.375	1723.282	1805.33	1853.755	1822.211	1739.404	1759.036	1834.276	1752.844	1756.867
	Rank	**1**	10	3	9	2	8	11	12	7	6	13	4	5
C17-F18	Mean	**1805.36**	2,877,257	11,923.85	5,735,592	11,111.67	12,127.7	23,449.89	21,073.62	20,025.71	29,687.97	9765.664	22,009.06	12,887.9
	Best	**1800.003**	148,282.3	4864.976	283,924	4174.747	7503.888	6480.271	8747.673	6354.336	24,138.12	6424.12	2888.071	3447.217
	Worst	**1820.451**	8,337,684	15,688.36	16,650,123	16,616.76	16,384.43	36,839.48	33,922.95	33,800.34	37,134.83	11,923.34	40,997.79	18,593.6
	Std	**10.95197**	4,127,821	5281.26	8,252,643	6158.261	4019.381	15,920.68	12,898.77	15,147.08	6506.053	2554.342	21,410.87	7199.978
	Median	**1800.492**	1,511,530	13,571.04	3,004,161	11,827.6	12,311.24	25,239.9	20,811.92	19,974.07	28,739.47	10,357.6	22,075.19	14,755.39
	Rank	**1**	12	4	13	3	5	10	8	7	11	2	9	6
C17-F19	Mean	**1900.445**	390,051.5	6740.448	708,553.5	5623.033	126,338	35,023.84	1914.851	5407.081	4715.34	40,679.05	25,099.05	6211.208
	Best	**1900.039**	25,762.01	2178.665	46,105.64	2320.659	1949.556	7697.99	1909.484	1944.955	2044.21	11,164.51	2629.429	2215.362
	Worst	**1901.559**	821,920.8	13,319.27	1,522,124	9466.148	252,369.2	64,127.87	1924.471	13,887.85	12,559.69	59,008.83	77,380.53	9935.635
	Std	**0.810364**	378,831.5	5895.722	724,659.8	3963.615	156,290.3	25,211.41	7.70358	6217.733	5691.553	23,319.35	38,358.7	3466.525
	Median	**1900.09**	356,261.6	5731.927	632,992.2	5352.663	125,516.6	34,134.74	1912.724	2897.759	2128.73	46,271.42	10,193.12	6346.918
	Rank	**1**	12	7	13	5	11	9	2	4	3	10	8	6
C17-F20	Mean	**2000.312**	2216.383	2171.683	2224.494	2092.931	2208.745	2207.956	2140.477	2171.033	2072.475	2255.372	2170.09	2050.542
	Best	**2000.312**	2165.24	2031.536	2165.608	2073.228	2107.405	2098.976	2047.247	2131.776	2061.375	2189.014	2145.806	2036.046
	Worst	**2000.312**	2283.827	2296.441	2280.283	2123.641	2323.021	2289.781	2249.067	2247.625	2082.964	2349.088	2202.147	2058.359
	Std	**0**	53.77105	129.6972	61.41781	23.50952	99.40453	99.264	90.18592	56.83593	9.850576	84.75517	30.47507	11.19601
	Median	**2000.312**	2208.232	2179.377	2226.043	2087.428	2202.278	2221.533	2132.797	2152.365	2072.78	2241.692	2166.204	2053.882
	Rank	**1**	11	8	12	4	10	9	5	7	3	13	6	2
C17-F21	Mean	**2200**	2292.224	2213.908	2267.615	2257.617	2326.087	2310.663	2253.534	2314.124	2300.419	2369.547	2319.669	2298.888
	Best	**2200**	2245.786	2204.158	2224.131	2255.109	2221.386	2218.528	2200.008	2309.885	2203.746	2351.941	2311.549	2226.752
	Worst	**2200**	2321.323	2239.299	2292.339	2260.172	2373.39	2355.179	2308.4	2319.13	2339.391	2387.001	2327.277	2333.787
	Std	**0**	37.4424	18.47872	32.82983	2.33168	77.27499	67.69154	67.27446	4.137785	70.64276	15.94573	8.418841	53.00527
	Median	**2200**	2300.893	2206.088	2276.995	2257.593	2354.786	2334.472	2252.863	2313.74	2329.27	2369.622	2319.925	2317.506
	Rank	**1**	6	2	5	4	12	9	3	10	8	13	11	7
C17-F22	Mean	2300.073	2701.027	2309.054	2920.786	2305.044	2717.182	2323.99	**2285.662**	2308.669	2319.729	2300.004	2313.376	2318.072
	Best	2300	2581.071	2304.399	2710.209	2300.951	2450.338	2319.288	**2228.873**	2301.277	2313.416	2300	2300.643	2315.149
	Worst	2300.29	2820.282	2311.236	3075.328	2309.437	2926.716	2331.697	2305.332	2322.59	2331.562	**2300.018**	2345.833	2322.574
	Std	0.157893	114.9318	3.421932	167.3244	3.892406	231.3835	6.033524	41.21509	10.66841	9.036157	**0.009658**	23.59913	3.452639
	Median	**2300**	2701.377	2310.292	2948.804	2304.894	2745.837	2322.488	2304.221	2305.405	2316.97	**2300**	2303.514	2317.282
	Rank	3	11	6	13	4	12	10	**1**	5	9	2	7	8
C17-F23	Mean	**2600.919**	2690.499	2642.521	2701.598	2614.457	2724.652	2649.231	2620.453	2613.868	2643	2793.704	2644.76	2656.735
	Best	**2600.003**	2655.377	2630.851	2672.314	2611.98	2634.724	2631.18	2607.24	2607.853	2632.022	2728.026	2637.475	2636.591
	Worst	**2602.87**	2710.595	2660.461	2742.593	2617.187	2769.526	2669.602	2632.143	2620.651	2652.382	2933.47	2656.811	2665.174
	Std	**1.436922**	28.38289	15.1817	35.76758	2.682389	66.28428	22.5696	11.80644	7.173201	9.830174	105.0806	9.521171	14.8412
	Median	**2600.403**	2698.012	2639.385	2695.743	2614.331	2747.18	2648.071	2621.215	2613.483	2643.799	2756.66	2642.376	2662.587
	Rank	**1**	10	5	11	3	12	8	4	2	6	13	7	9
C17-F24	Mean	**2630.488**	2788.317	2768.95	2852.038	2630.654	2668.659	2761.886	2683.834	2749.937	2757.061	2748.612	2766.885	2724.025
	Best	2516.677	2745.688	2736.941	2825.654	2617.619	2523.669	2736.445	**2501.19**	2726.533	2745.923	2502.655	2755.88	2536.059
	Worst	2732.32	2856.987	2792.846	2913.462	**2636.807**	2812.827	2792.301	2759.842	2766.303	2767.028	2899.211	2786.936	2811.901
	Std	126.7883	57.95513	27.89637	44.82121	**9.562201**	168.3131	25.02914	133.1237	19.19886	10.60604	186.1525	15.09998	137.7502
	Median	2636.477	2775.296	2773.006	2834.518	**2634.095**	2669.069	2759.399	2737.151	2753.457	2757.647	2796.292	2762.363	2774.07
	Rank	**1**	12	11	13	2	3	9	4	7	8	6	10	5
C17-F25	Mean	2932.639	3139.215	2913.317	3279.873	2917.765	3135.421	**2907.315**	2922.007	2938.751	2933.537	2922.179	2923.252	2952.419
	Best	2898.047	3067.756	2899.104	3210.527	2913.428	2905.513	**2763.108**	2900.572	2921.074	2915.907	2902.261	2898.673	2938.562
	Worst	2945.793	3299.244	2949.092	3356.883	**2923.18**	3664.488	2959.682	2943.722	2945.915	2952.388	2943.394	2946.56	2962.947
	Std	25.12878	118.4951	26.01647	65.88501	**4.450199**	388.1053	104.7267	26.80868	12.86286	21.96492	24.96584	28.66542	11.32013
	Median	2943.359	3094.931	**2902.537**	3276.041	2917.225	2985.842	2953.235	2921.867	2944.007	2932.927	2921.531	2923.888	2954.084
	Rank	7	12	2	13	3	11	**1**	4	9	8	5	6	10
C17-F26	Mean	2900	3563.475	2980.612	3764.465	3012.721	3627.651	3185.71	2900.149	3268.794	3209.548	3870.812	2904.098	**2897.19**
	Best	2900	3234.392	2806.117	3437.518	2892.04	3146.539	2927.473	2900.114	2969.885	2912.169	2806.117	2806.117	**2705.581**
	Worst	**2900**	3783.253	3159.257	4104.73	3297.318	4282.661	3600.731	2900.195	3916.767	3885.046	4362.862	3010.276	3111.603
	Std	**4.04E-13**	264.6671	219.3124	313.0895	207.4462	604.6037	320.3471	0.039308	474.4582	493.3205	785.0288	90.85344	223.8032
	Median	2900	3618.128	2978.537	3757.805	2930.763	3540.701	3107.318	2900.144	3094.262	3020.488	4157.136	2900	**2885.789**
	Rank	2	10	5	12	6	11	7	3	9	8	13	4	**1**
C17-F27	Mean	**3089.518**	3211.093	3120.294	3232.474	3104.836	3180.387	3195.929	3091.648	3116.364	3115.336	3227.33	3136.519	3160.678
	Best	**3089.518**	3162.744	3095.362	3127.574	3092.27	3102.552	3179.884	3089.712	3094.484	3095.441	3215.052	3097.168	3119.628
	Worst	**3089.518**	3293.2	3181.796	3426.382	3134.233	3223.046	3207.804	3095.016	3177.613	3172.045	3249.089	3184.29	3220.171
	Std	**2.86E-13**	61.76378	44.75181	144.0715	21.48364	59.41073	12.68162	2.715154	44.48251	41.1546	16.48412	39.87289	46.26169
	Median	**3089.518**	3194.214	3102.009	3187.971	3096.421	3197.974	3198.014	3090.932	3096.68	3096.929	3222.589	3132.309	3151.456
	Rank	**1**	11	6	13	3	9	10	2	5	4	12	7	8
C17-F28	Mean	**3100**	3597.002	3237.24	3784.862	3219.529	3590.319	3288.358	3239.881	3346.978	3326.958	3453.664	3307.374	3247.568
	Best	**3100**	3551.344	**3100**	3701.869	3167.488	3415.098	3153.13	3100.125	3195.479	3214.92	3440.292	3177.754	3145.253
	Worst	**3100**	3629.166	3392.748	3844.77	3244.627	3801.261	3393.263	3392.749	3414.627	3392.992	3472.246	3392.965	3516.826
	Std	**0**	37.84994	140.9244	72.18444	38.85546	217.9689	134.3096	175.9315	110.7805	92.50073	16.11354	106.1885	196.1013
	Median	**3100**	3603.75	3228.106	3796.405	3232.999	3572.459	3303.518	3233.325	3388.902	3349.961	3451.059	3329.389	3164.097
	Rank	**1**	12	3	13	2	11	6	4	9	8	10	7	5
C17-F29	Mean	**3132.241**	3341.543	3286.029	3377.573	3203.813	3237.271	3351.007	3203.391	3266.497	3213.443	3347.969	3267.377	3238.278
	Best	**3130.076**	3320.37	3211.196	3305.533	3166.325	3166.435	3236.743	3142.631	3190.458	3166.015	3234.677	3168.273	3189.013
	Worst	**3134.841**	3357.583	3367.588	3445.427	3245.657	3308.027	3499.364	3288.015	3381.802	3236.348	3639.853	3351.013	3287.876
	Std	**2.701544**	16.93059	87.65176	78.4692	38.01024	63.06594	119.873	67.0027	99.05148	35.89995	212.6507	90.30554	45.26891
	Median	**3132.023**	3344.109	3282.666	3379.666	3201.635	3237.31	3333.962	3191.459	3246.864	3225.705	3258.672	3275.111	3238.111
	Rank	**1**	10	9	13	3	5	12	2	7	4	11	8	6
C17-F30	Mean	**3418.734**	2,270,202	296,246.2	3,694,956	416,890.8	617,692.3	997,284.6	304,439.1	940,663.9	60,930.4	786,751.4	389,254.4	1,535,217
	Best	**3394.682**	1,673,245	105,205.9	831,886.7	15,996.07	112,875.5	4471.247	7460.759	33,746.81	29,426.72	604,858.7	6409.895	528,520.6
	Worst	**3442.907**	3,240,778	771,775.4	5,836,135	615,301.1	1,306,089	3,764,900	1,160,773	1,361,385	102,270.8	1,004,710	771,812.2	3,497,487
	Std	**30.22288**	740,116	345,952.7	2,280,306	296,209	551,800.6	2,010,505	621,503.5	678,871.6	38,719.09	180,833.5	480,108.6	1,523,052
	Median	**3418.673**	2,083,393	154,001.9	4,055,902	518,132.9	525,902.3	109,883.7	24,761.28	1,183,762	56,012.05	768,718.5	389,397.8	1,057,430
	Rank	**1**	12	3	13	6	7	10	4	9	2	8	5	11
Sum rank		**38**	318	177	350	106	286	239	116	188	191	238	183	197
Mean rank		**1.310345**	10.96552	6.103448	12.06897	3.655172	9.862069	8.241379	4	6.482759	6.586207	8.206897	6.310345	6.793103
Total rank		**1**	12	4	13	2	11	10	3	6	7	9	5	8

**Table 4 biomimetics-09-00137-t004:** Wilcoxon signed-rank test results.

Compared Algorithm	Objective Function Type
	CEC 2017
BOA vs. WSO	1.97E-21
BOA vs. AVOA	3.77E-19
BOA vs. RSA	1.97E-21
BOA vs. MPA	2.00E-18
BOA vs. TSA	9.50E-21
BOA vs. WOA	9.50E-21
BOA vs. MVO	9.03E-19
BOA vs. GWO	5.23E-21
BOA vs. TLBO	3.69E-21
BOA vs. GSA	1.60E-18
BOA vs. PSO	1.54E-19
BOA vs. GA	2.71E-19

**Table 5 biomimetics-09-00137-t005:** Optimization results of the CEC 2011 test suite.

		BOA	WSO	AVOA	RSA	MPA	TSA	WOA	MVO	GWO	TLBO	GSA	PSO	GA
C11-F1	Mean	**5.920103**	18.3658	13.36522	22.89003	7.662645	19.13919	13.67372	14.47026	11.14773	19.17297	22.59503	18.65142	24.38586
	Best	**2E-10**	16.19408	9.345854	21.25619	0.392096	18.18294	8.678559	11.68925	1.176288	17.33253	20.70269	11.03757	23.48431
	Worst	**12.30606**	21.05886	17.2206	25.24276	12.70674	20.36399	17.69423	16.92367	18.35369	21.06058	23.95826	24.97436	26.34835
	Std	7.196379	2.475039	4.55974	1.997624	5.912131	**1.046045**	4.319552	2.53189	7.606047	1.629171	1.471501	6.959545	1.418008
	Median	**5.687176**	18.10514	13.44722	22.53059	8.775874	19.00491	14.16104	14.63406	12.53047	19.14939	22.85957	19.29688	23.85539
	Rank	**1**	7	4	12	2	9	5	6	3	10	11	8	13
C11-F2	Mean	**−26.3179**	−13.8385	−20.8022	−10.8912	−25.0347	−10.5975	−18.2689	−7.99707	−22.4684	−10.1862	−15.0459	−22.5196	−12.3132
	Best	**−27.0676**	−15.2725	−21.3417	−11.3546	−25.6818	−14.5354	−21.8709	−10.1042	−24.6938	−11.4086	−20.3083	−23.9159	−14.7797
	Worst	**−25.4328**	−12.5692	−20.0389	−10.403	−23.6635	−8.29419	−14.0672	−6.404	−18.7253	−9.11933	−10.7963	−20.0088	−10.5245
	Std	0.738935	1.450672	0.613406	**0.51655**	0.987822	3.104455	4.221461	1.682368	2.761006	0.998848	4.557654	1.808243	2.093573
	Median	**−26.3856**	−13.7562	−20.914	−10.9036	−25.3968	−9.78013	−18.5688	−7.74007	−23.2272	−10.1085	−14.5394	−23.0768	−11.9743
	Rank	**1**	8	5	10	2	11	6	13	4	12	7	3	9
C11-F4	Mean	**1.15E-05**	**1.15E-05**	**1.15E-05**	**1.15E-05**	**1.15E-05**	**1.15E-05**	**1.15E-05**	**1.15E-05**	**1.15E-05**	**1.15E-05**	**1.15E-05**	**1.15E-05**	**1.15E-05**
	Best	**1.15E-05**	**1.15E-05**	**1.15E-05**	**1.15E-05**	**1.15E-05**	**1.15E-05**	**1.15E-05**	**1.15E-05**	**1.15E-05**	**1.15E-05**	**1.15E-05**	**1.15E-05**	**1.15E-05**
	Worst	**1.15E-05**	**1.15E-05**	**1.15E-05**	**1.15E-05**	**1.15E-05**	**1.15E-05**	**1.15E-05**	**1.15E-05**	**1.15E-05**	**1.15E-05**	**1.15E-05**	**1.15E-05**	**1.15E-05**
	Std	2E-19	2.29E-11	2.63E-09	5.16E-11	1.28E-15	2.46E-14	6.39E-19	1.03E-12	3.85E-15	8.1E-14	2.07E-19	**6.03E-20**	2.85E-18
	Median	**1.15E-05**	**1.15E-05**	**1.15E-05**	**1.15E-05**	**1.15E-05**	**1.15E-05**	**1.15E-05**	**1.15E-05**	**1.15E-05**	**1.15E-05**	**1.15E-05**	**1.15E-05**	**1.15E-05**
	Rank	**1**	11	13	12	6	8	4	10	7	9	3	2	5
C11-F4	Mean	**0**	**0**	**0**	**0**	**0**	**0**	**0**	**0**	**0**	**0**	**0**	**0**	**0**
	Best	**0**	**0**	**0**	**0**	**0**	**0**	**0**	**0**	**0**	**0**	**0**	**0**	**0**
	Worst	**0**	**0**	**0**	**0**	**0**	**0**	**0**	**0**	**0**	**0**	**0**	**0**	**0**
	Std	**0**	**0**	**0**	**0**	**0**	**0**	**0**	**0**	**0**	**0**	**0**	**0**	**0**
	Median	**0**	**0**	**0**	**0**	**0**	**0**	**0**	**0**	**0**	**0**	**0**	**0**	**0**
	Rank	**1**	**1**	**1**	**1**	**1**	**1**	**1**	**1**	**1**	**1**	**1**	**1**	**1**
C11-F5	Mean	**−34.1274**	−24.4632	−27.8918	−19.4252	−33.246	−26.8779	−27.396	−26.7327	−31.4832	−9.86988	−27.1031	−7.61915	−8.51327
	Best	**−34.7494**	−25.646	−28.9861	−21.6476	−33.8296	−31.4861	−27.5536	−31.6474	−34.2023	−12.1106	−31.4408	−11.318	−10.0071
	Worst	**−33.3862**	−23.4985	−27.4102	−16.9763	−31.872	−21.3206	−27.0038	−24.211	−27.3029	−8.15331	−23.8495	−5.81752	−6.75594
	Std	0.589989	0.989457	0.779829	2.589014	0.96747	4.401075	**0.27604**	3.659252	3.096495	1.770206	3.496342	2.723063	1.505232
	Median	**−34.1871**	−24.3541	−27.5855	−19.5384	−33.6412	−27.3525	−27.5132	−25.5362	−32.2137	−9.60779	−26.561	−6.67054	−8.64499
	Rank	**1**	9	4	10	2	7	5	8	3	11	6	13	12
C11-F6	Mean	**−24.1119**	−13.6555	−18.847	−12.6221	−22.5646	−6.92131	−19.8051	−8.96998	−19.4699	−1.47492	−21.8112	−2.37518	−3.31781
	Best	**−27.4298**	−14.2972	−20.1867	−13.3562	−25.6947	−16.2979	−22.9894	−17.2225	−22.2246	−1.67787	−26.7438	−5.2789	−8.77883
	Worst	**−23.0059**	−13.3329	−17.0321	−11.6073	−21.2709	−3.56713	−12.5777	−1.40727	−17.8008	−1.40727	−17.4414	−1.40727	−1.40727
	Std	2.324951	0.457816	1.549487	0.868125	2.224729	6.579278	5.181926	9.043209	2.229771	**0.142217**	4.201888	2.034739	3.829068
	Median	**−23.0059**	−13.496	−19.0847	−12.7624	−21.6463	−3.9101	−21.8266	−8.62506	−18.9272	−1.40727	−21.5297	−1.40727	−1.54257
	Rank	**1**	7	6	8	2	10	4	9	5	13	3	12	11
C11-F7	Mean	**0.860699**	1.630336	1.297839	1.954375	0.931882	1.316263	1.772373	0.881731	1.074249	1.746626	1.086692	1.132047	1.769103
	Best	**0.582266**	1.576312	1.151917	1.700334	0.76299	1.146539	1.652665	0.811995	0.807742	1.544826	0.894221	0.827051	1.374402
	Worst	1.025027	1.738867	1.439848	2.140834	1.012213	1.688543	1.946221	**0.958962**	1.308118	1.888265	1.294407	1.382585	1.976253
	Std	0.211503	0.078973	0.161247	0.194066	0.121385	0.263306	0.131457	**0.078602**	0.217578	0.157566	0.191475	0.30478	0.286366
	Median	0.91775	1.603082	1.299796	1.988165	0.976163	1.214985	1.745302	**0.877984**	1.090569	1.776707	1.079071	1.159275	1.862878
	Rank	**1**	9	7	13	3	8	12	2	4	10	5	6	11
C11-F8	Mean	**220**	287.3329	241.2176	329.2094	222.5348	258.6563	267.7395	224.2247	227.6045	224.2247	247.3067	478.5826	222.5818
	Best	**220**	259.8815	223.7553	286.7508	**220**	**220**	246.1934	**220**	**220**	**220**	**220**	249.1037	**220**
	Worst	**220**	323.3464	258.6798	375.4703	225.0697	359.4163	315.479	236.8989	235.209	236.8989	296.0452	582.7551	230.3271
	Std	**0**	29.20661	15.7971	38.25039	3.076552	71.0069	33.71364	8.881242	9.229657	8.881242	37.90414	165.984	5.427425
	Median	**220**	283.0518	241.2176	327.3083	222.5348	227.6045	254.6428	**220**	227.6045	**220**	236.5908	541.2357	**220**
	Rank	**1**	10	6	11	2	8	9	4	5	4	7	12	3
C11-F9	Mean	**8789.286**	577,676.6	392,136.4	1,101,221	20,625	68,367.45	388,333.6	138,018.7	44,296.44	423,556.8	853,670.6	1,122,447	2,014,720
	Best	**5457.674**	385,879.1	346,854.4	718,850	11,156.31	49,109.4	214,821.2	78,052.73	18,844.04	350,374.3	730,315.2	900,965.2	1,930,602
	Worst	**14,042.29**	663,904.2	422,189.5	1,292,119	29,564.96	86,903.2	658,126.8	208,994.3	77,779.54	543,358.7	919,076.4	1,374,733	2,132,738
	Std	**3889.181**	137,784.3	34,745.74	273,210.9	8526.786	16,900.77	212,512.5	56,771.19	26,179.21	89,242.55	88,329.7	266,242.6	104,535.7
	Median	**7828.591**	630,461.6	399,750.8	1,196,958	20,889.37	68,728.59	340,193.1	132,513.8	40,281.09	400,247.2	882,645.4	1,107,045	1,997,771
	Rank	**1**	9	7	11	2	4	6	5	3	8	10	12	13
C11-F10	Mean	**−21.4889**	−13.701	−16.7579	−11.948	−18.9391	−14.1356	−12.564	−14.4587	−13.8394	−10.9206	−12.8545	−11.024	−10.7178
	Best	**−21.8299**	−14.9479	−16.9487	−12.3376	−19.3304	−18.7599	−13.2827	−21.164	−14.3313	−11.0122	−13.3834	−11.0692	−10.7552
	Worst	**−20.7878**	−13.0631	−16.3898	−11.6609	−18.554	−11.6783	−12.0574	−11.0939	−12.6316	−10.8456	−12.0435	−11.0025	−10.6622
	Std	0.498616	0.900402	0.26946	0.303776	0.42259	3.348581	0.542185	4.773199	0.85429	0.076044	0.682286	**0.032754**	0.042033
	Median	**−21.669**	−13.3964	−16.8466	−11.8967	−18.936	−13.0521	−12.4579	−12.7884	−14.1973	−10.9123	−12.9956	−11.0121	−10.7269
	Rank	**1**	7	3	10	2	5	9	4	6	12	8	11	13
C11-F11	Mean	**571,712.3**	5,990,542	1,005,937	9,157,615	1,697,924	6,138,051	1,238,362	1,334,588	3,950,347	5,376,638	1,441,156	5,388,114	6,322,408
	Best	**260,837.9**	5,713,697	790,621.8	8,861,897	1,582,223	5,108,653	1,126,726	609,492	3,753,138	5,357,622	1,292,303	5,372,054	6,288,702
	Worst	**828,560.9**	6,367,119	1,183,769	9,345,099	1,828,453	7,420,263	1,396,911	2,812,910	4,332,212	5,392,350	1,619,216	5,407,480	6,399,460
	Std	260,922.1	315,719.1	180,244.7	217,528.6	122,969.1	1,003,762	119,865	1,050,363	273,781	**15,960.55**	141,558.1	16,144.36	55,013.89
	Median	**598,725.2**	5,940,676	1,024,678	9,211,733	1,690,509	6,011,644	1,214,905	957,974.7	3,858,018	5,378,291	1,426,551	5,386,462	6,300,734
	Rank	**1**	10	2	13	6	11	3	4	7	8	5	9	12
C11-F12	Mean	**1,199,805**	8,648,465	3,453,394	13,667,035	1,277,229	5,166,688	5,980,120	1,332,009	1,432,076	14,799,181	5,954,053	2,353,727	14,965,863
	Best	**1,155,937**	8,290,445	3,348,644	12,692,541	1,199,938	4,885,754	5,545,813	1,174,160	1,263,209	13,925,728	5,653,564	2,175,729	14,835,162
	Worst	**1,249,353**	8,965,661	3,521,972	14,523,280	1,357,501	5,316,545	6,197,392	1,482,783	1,573,795	15,476,182	6,170,450	2,571,333	15,100,976
	Std	**47,157.58**	294,829.2	79,624.52	789,278.6	72,305.18	210,271.9	315,606	132,531.8	135,346.5	683,800.8	234,270.3	171,464.7	114,159.5
	Median	**1,196,965**	8,668,878	3,471,481	13,726,159	1,275,738	5,232,227	6,088,637	1,335,547	1,445,649	14,897,407	5,996,098	2,333,922	14,963,658
	Rank	**1**	10	6	11	2	7	9	3	4	12	8	5	13
C11-F13	Mean	**15,444.2**	15,872.72	15,448.13	16,341.93	15,463.86	15,491.87	15,538.8	15,510.22	15,503.18	15,951.03	132,457.9	15,492.53	30,533.99
	Best	**15,444.19**	15,680.46	15,447.1	15,910.59	15,461.47	15,481.59	15,493.51	15,489.25	15,496	15,634.2	95,600.27	15,474.67	15,461.05
	Worst	**15,444.21**	16,338.51	15,449.28	17,413.55	15,468	15,504.94	15,599.85	15,550.13	15,515.6	16,534.68	182,449.7	15,530.38	75,388.3
	Std	**0.009091**	329.5679	0.965292	757.1036	3.04201	12.13496	52.00076	29.66598	9.128434	428.3972	41,099.67	26.81114	31,431.06
	Median	**15,444.2**	15,735.95	15,448.08	16,021.78	15,462.99	15,490.47	15,530.92	15,500.75	15,500.56	15,817.61	125,890.8	15,482.53	15,643.31
	Rank	**1**	9	2	11	3	4	8	7	6	10	13	5	12
C11-F14	Mean	**18,295.35**	115,938.6	18,527.59	236,838	18,619.25	19,576.66	19,259.86	19,460.62	19,267.06	321,549.6	19,121.96	19,155.72	19,142.49
	Best	**18,241.58**	88,038.86	18,409.22	174,242.1	18,533.36	19,310.27	19,102.02	19,357.41	19,116.91	30,687.52	18,822.08	18,985.66	18,847.36
	Worst	**18,388.08**	162,435.6	18,626.34	341,619.3	18,694.96	20,146.79	19,385.69	19,546.24	19,460.5	621,276.2	19,341.44	19,304.28	19,454.19
	Std	**71.59938**	34,977.03	107.5807	78,804.57	73.40623	403.5767	136.4873	83.23058	159.3053	298,019.5	235.5233	137.423	260.6657
	Median	**18,275.87**	106,639.9	18,537.4	215,745.3	18,624.35	19,424.79	19,275.87	19,469.41	19,245.42	317,117.3	19,162.17	19,166.47	19,134.21
	Rank	**1**	11	2	12	3	10	7	9	8	13	4	6	5
C11-F15	Mean	**32,883.58**	940,841.1	110,645.3	1,985,231	32,950.76	55,363.33	225,557.8	33,108.08	33,085.12	15,996,211	309,756.9	33,303.15	8,231,969
	Best	**32,782.17**	387,070.3	43,553.37	829,509.7	32,873.13	33,051.11	33,017.86	33,024.19	33,048.03	3,350,790	274,007.7	33,293.78	3,746,244
	Worst	**32,956.46**	2,367,653	185,709.1	5,184,170	33,020.99	121,996.9	323,378.9	33,169.01	33,150.04	23,854,606	334,233.8	33,312.43	14,108,888
	Std	76.94696	1,003,433	80,304.62	2,245,091	63.64504	46,692.58	137,782.2	67.04471	48.92391	9,799,500	29,448.29	**8.071454**	4,994,206
	Median	**32,897.86**	504,320.5	106,659.4	963,621.3	32,954.47	33,202.65	272,917.2	33,119.57	33,071.21	18,389,724	315,393.1	33,303.19	7,536,372
	Rank	**1**	10	7	11	2	6	8	4	3	13	9	5	12
C11-F16	Mean	**133,550**	975,518.1	135,237.7	2,017,200	137,810.6	145,558	142,484.5	142,115.8	146,331.8	92,220,903	19,417,718	82,541,543	79,253,422
	Best	**131,374.2**	294,692.9	133,737.1	486,424.7	135,730.3	142,830.3	136,399.6	133,165.5	143,684	89,866,744	9,860,015	68,276,866	64,052,715
	Worst	136,310.8	2,311,282	**135,911.9**	5,020,540	141,530.1	147,640.7	147,906.6	151,316.2	151,968.7	94,876,220	35,135,215	98,635,651	1.01E+08
	Std	2392.2	953,315.1	**1067.526**	2,143,364	2722.974	2482.867	5054.284	8022.252	4005.458	2,206,781	11,487,547	13,754,292	16,662,950
	Median	**133,257.5**	648,048.9	135,650.9	1,280,917	136,991	145,880.4	142,816	141,990.7	144,837.2	92,070,325	16,337,822	81,626,828	75,794,234
	Rank	**1**	8	2	9	3	6	5	4	7	13	10	12	11
C11-F17	Mean	**1,926,615**	9.3E+09	2.4E+09	1.61E+10	2,304,570	1.33E+09	1.01E+10	3,156,432	3,060,480	2.31E+10	1.16E+10	2.16E+10	2.27E+10
	Best	**1,916,953**	7.92E+09	2.18E+09	1.16E+10	1,958,863	1.1E+09	7.18E+09	2,310,026	2,042,683	2.23E+10	1.02E+10	1.91E+10	2.12E+10
	Worst	**1,942,685**	1.03E+10	2.63E+09	1.97E+10	2,944,065	1.52E+09	1.34E+10	3,810,703	4,991,555	2.42E+10	1.23E+10	2.5E+10	2.56E+10
	Std	**12,003.53**	1.11E+09	2.07E+08	3.66E+09	464,536.3	2.29E+08	2.74E+09	728,055.4	1,396,444	8.21E+08	9.97E+08	2.8E+09	2.11E+09
	Median	**1,923,412**	9.48E+09	2.4E+09	1.66E+10	2,157,676	1.35E+09	9.84E+09	3,252,498	2,603,840	2.31E+10	1.2E+10	2.12E+10	2.2E+10
	Rank	**1**	7	6	10	2	5	8	4	3	13	9	11	12
C11-F18	Mean	**942,057.5**	57,009,339	6,765,597	1.23E+08	972,857.5	2,091,646	9,903,113	989,837.6	1,034,458	32,127,447	11,509,537	1.4E+08	1.19E+08
	Best	**938,416.2**	39,198,119	4,054,026	84,796,562	950,200.1	1,824,897	4,240,455	964,629.7	967,922.6	25,456,959	8,580,056	1.17E+08	1.14E+08
	Worst	**944,706.9**	64,852,362	11,629,143	1.4E+08	1,033,181	2,447,367	17,411,962	1,001,598	1,210,129	34,756,120	14,528,317	1.55E+08	1.23E+08
	Std	**2774.139**	12,627,720	3,707,867	27,267,025	42,401.09	315,442.7	5,845,999	17,904.23	123,353.4	4,693,486	2,793,344	17,827,026	3,754,354
	Median	**942,553.5**	61,993,438	5,689,610	1.33E+08	954,024.6	2,047,161	8,980,017	996,561.4	979,889.4	34,148,355	11,464,888	1.43E+08	1.19E+08
	Rank	**1**	10	6	12	2	5	7	3	4	9	8	13	11
C11-F19	Mean	**1,025,341**	56,112,190	6,867,491	1.2E+08	1,142,037	2,517,276	10,562,701	1,493,568	1,375,430	36,886,795	6,463,721	1.79E+08	1.19E+08
	Best	**967,927.7**	47,875,067	6,260,767	1.04E+08	1,070,955	2,270,154	2,100,547	1,134,176	1,241,437	25,821,500	2,432,836	1.63E+08	1.16E+08
	Worst	**1,167,142**	71,354,860	8,329,100	1.51E+08	1,297,605	2,980,670	19,167,084	1,996,238	1,558,531	46,022,778	8,501,596	2.07E+08	1.23E+08
	Std	**99,675.04**	11,137,193	1,031,447	23,175,948	110,088.2	333,115.4	8,443,038	380,000.6	139,424	9,198,239	2,895,913	20,333,979	2,808,423
	Median	**983,146.6**	52,609,416	6,440,048	1.13E+08	1,099,794	2,409,140	10,491,585	1,421,928	1,350,877	37,851,450	7,460,226	1.73E+08	1.19E+08
	Rank	**1**	10	7	12	2	5	8	4	3	9	6	13	11
C11-F20	Mean	**941,250.4**	59,668,611	6,078,019	1.3E+08	961,061.9	1,859,713	7,518,662	973,995.9	1,000,685	35,832,144	14,770,447	1.65E+08	1.2E+08
	Best	**936,143.2**	52,493,138	5,354,905	1.14E+08	957,468.7	1,668,863	7,081,933	963,584.3	978,672.9	35,045,378	9,799,344	1.51E+08	1.14E+08
	Worst	**946,866.6**	70,665,769	6,851,445	1.55E+08	963,379.7	2,176,763	8,101,614	985,820.9	1,017,853	36,682,873	22,883,459	1.79E+08	1.24E+08
	Std	5013.552	8,139,325	652,894.4	18,267,776	**2670.851**	253,461.5	458,251.8	10,320.56	17,752.81	715,894.6	6,010,182	16,641,832	4,510,598
	Median	**940,995.9**	57,757,768	6,052,862	1.26E+08	961,699.6	1,796,612	7,445,551	973,289.1	1,003,107	35,800,163	13,199,493	1.65E+08	1.2E+08
	Rank	**1**	10	6	12	2	5	7	3	4	9	8	13	11
C11-F21	Mean	**12.71443**	51.66477	21.98088	78.87305	16.0572	30.47192	39.77733	28.09829	22.74275	104.0164	41.7608	109.2417	105.994
	Best	**9.974206**	42.34744	20.68456	58.47624	13.90111	27.01848	36.26677	24.84148	20.91572	49.63025	36.61142	94.29664	60.45563
	Worst	**14.97499**	61.65781	23.79304	99.26551	18.3479	32.19298	44.13805	31.18011	25.09372	153.4827	44.73996	121.6858	129.6288
	Std	2.412667	8.750969	**1.396524**	18.93558	2.173182	2.482172	3.658371	3.733097	1.924417	44.71405	3.837542	14.11278	33.81205
	Median	**12.95425**	51.32692	21.72295	78.87522	15.98989	31.33811	39.35224	28.18579	22.48077	106.4764	42.84591	110.4922	116.9458
	Rank	**1**	9	3	10	2	6	7	5	4	11	8	13	12
C11-F22	Mean	**16.12513**	47.98	27.87551	65.28358	19.19349	32.74598	47.48138	32.90803	25.33021	105.8946	47.85216	110.0809	95.48389
	Best	**11.50133**	41.46275	22.5794	46.99819	16.36546	28.71428	41.10402	25.17317	24.09788	68.38003	39.74339	92.35913	94.65695
	Worst	**19.55286**	53.63684	33.24577	75.15155	21.32566	35.26266	52.30679	38.08284	26.25798	125.4222	57.19442	121.509	97.01353
	Std	4.197797	5.483258	5.290419	13.16113	2.482115	2.995329	5.301012	6.106807	**1.080977**	26.96416	7.537477	13.85053	1.132853
	Median	**16.72317**	48.41021	27.83842	69.4923	19.54142	33.50349	48.25735	34.18806	25.48249	114.8881	47.23541	113.2277	95.13255
	Rank	**1**	9	4	10	2	5	7	6	3	12	8	13	11
Sum rank		**22**	191	109	231	55	146	145	118	97	222	157	198	224
Mean rank		**1**	8.681818	4.954545	10.5	2.5	6.636364	6.590909	5.363636	4.409091	10.09091	7.136364	9	10.18182
Total rank		**1**	9	4	13	2	7	6	5	3	11	8	10	12
Wilcoxon: *p*-value			4.38E-12	7.75E-15	1.56E-15	0.001746142	4.89E-15	5.25E-15	1.60E-11	1.92E-12	3.34E-15	8.03E-15	1.56E-15	2.28E-15

**Table 6 biomimetics-09-00137-t006:** Performance of optimization algorithms on the pressure vessel design problem.

Algorithm	Optimal Variables				Optimal Cost
	$T_{s}$	$T_{h}$	*R*	*L*	
BOA	0.7781685	0.3846492	40.319615	200	5885.3263
WSO	0.7781685	0.3846492	40.319615	200	5885.3322
AVOA	0.7781902	0.3846599	40.320737	199.98436	5885.3693
RSA	0.8538832	0.4168324	40.384824	200	6547.2433
MPA	0.7781685	0.3846492	40.319615	200	5885.3322
TSA	0.7797576	0.3858656	40.396539	200	5913.0266
WOA	0.8128457	0.5410128	40.396424	198.93351	6581.148
MVO	0.8182022	0.4061992	42.352706	173.53515	5968.7271
GWO	0.7784539	0.3856252	40.32716	199.94288	5890.2366
TLBO	1.1978845	1.2639942	61.056149	91.741579	14,709.571
GSA	0.957018	0.4737273	49.581732	144.99985	7674.4943
PSO	1.276768	2.3221525	50.647017	110.15343	17,231.342
GA	1.1434315	0.7799385	54.784767	96.514991	9745.9413

**Table 7 biomimetics-09-00137-t007:** Statistical results of optimization algorithms on the pressure vessel design problem.

Algorithm	Mean	Best	Worst	Std	Median	Rank
BOA	5885.3263	5885.3263	5885.3263	2.32E-08	5885.3263	1
WSO	5907.011	5885.3322	6094.606	53.104713	5885.3322	3
AVOA	6417.9542	5885.3693	7301.8987	485.20827	6249.9206	5
RSA	12,102.458	6547.2433	20,969.982	3923.6076	11,268.435	9
MPA	5885.3322	5885.3322	5885.3322	3.91E-06	5885.3322	2
TSA	6259.4568	5913.0266	7323.2568	391.18418	6101.1237	6
WOA	7978.5279	6581.148	12,433.242	1390.3395	7795.4774	8
MVO	6576.3819	5968.7271	7273.5044	448.12807	6572.6459	7
GWO	5945.5243	5890.2366	6636.6942	163.66397	5901.7573	4
TLBO	39,032.934	14,709.571	69,674.574	15,506.903	38,454.338	12
GSA	24,592.049	7674.4943	39,531.957	8743.4829	26,413.075	10
PSO	41,176.997	17,231.342	89,983.875	18,842.417	38,677.472	13
GA	29,575.451	9745.9413	60,485.672	14,026.27	26,621.057	11

**Table 8 biomimetics-09-00137-t008:** Performance of optimization algorithms on the speed reducer design problem.

Algorithm	Optimal Variables							Optimal Cost
	b	M	p	$l_{1}$	$l_{2}$	$d_{1}$	$d_{2}$	
BOA	3.5	0.7	17	7.3	7.8	3.3502147	5.2866832	2996.3482
WSO	3.5000005	0.7	17	7.3000099	7.8000004	3.3502148	5.2866833	2996.3483
AVOA	3.5	0.7	17	7.3000007	7.8	3.3502147	5.2866832	2996.3482
RSA	3.5922092	0.7	17	8.222092	8.261046	3.3556658	5.4833809	3182.9113
MPA	3.5	0.7	17	7.3	7.8	3.3502147	5.2866832	2996.3482
TSA	3.5129039	0.7	17	7.3	8.261046	3.3505407	5.2902177	3013.8833
WOA	3.587509	0.7	17	7.3	8.0094193	3.3616163	5.2867558	3038.2679
MVO	3.5022528	0.7	17	7.3	8.069157	3.3696027	5.2868819	3008.2394
GWO	3.5006415	0.7	17	7.3051454	7.8	3.3639533	5.2888109	3001.5161
TLBO	3.556121	0.703999	26.327655	8.1017162	8.1453492	3.6635667	5.3393802	5271.2441
GSA	3.5229197	0.7027544	17.369301	7.8207543	7.8896487	3.4088005	5.3859782	3169.7986
PSO	3.5081873	0.700072	18.096129	7.3990809	7.8680597	3.5955569	5.3440486	3302.6701
GA	3.5780478	0.7055678	17.814174	7.742773	7.8558683	3.7017132	5.3463595	3.35E+03

**Table 9 biomimetics-09-00137-t009:** Statistical results of optimization algorithms on the speed reducer design problem.

Algorithm	Mean	Best	Worst	Std	Median	Rank
BOA	2996.3482	2996.3482	2996.3482	9.33E-13	2996.3482	1
WSO	2996.6318	2996.3483	2998.8003	0.5851051	2996.3644	3
AVOA	3000.8579	2996.3482	3011.0816	3.9697349	3000.7583	4
RSA	3276.9058	3182.9113	3335.2402	57.540902	3291.7902	9
MPA	2996.3482	2996.3482	2996.3482	3.19E-06	2996.3482	2
TSA	3032.1482	3013.8833	3045.8852	10.144159	3033.9369	7
WOA	3150.1207	3038.2679	3445.3098	106.34463	3116.768	8
MVO	3029.8375	3008.2394	3070.2104	13.263239	3030.2775	6
GWO	3004.6252	3001.5161	3010.5926	2.5083807	3004.107	5
TLBO	6.958E+13	5271.2441	5.037E+14	1.158E+14	2.725E+13	12
GSA	3454.8489	3169.7986	4076.1493	262.31973	3325.1431	10
PSO	1.027E+14	3302.6701	5.202E+14	1.24E+14	7.345E+13	13
GA	4.944E+13	3347.0081	3.191E+14	7.789E+13	1.981E+13	11

**Table 10 biomimetics-09-00137-t010:** Performance of optimization algorithms on the welded beam design problem.

Algorithm	Optimal Variables				Optimal Cost
	*h*	*l*	*t*	*b*	
BOA	0.2057296	3.4704887	9.0366239	0.2057296	1.7246798
WSO	0.2057296	3.4704887	9.0366239	0.2057296	1.7248523
AVOA	0.2049647	3.4870781	9.0365172	0.2057345	1.7259197
RSA	0.1966937	3.534683	9.9249453	0.2177987	1.9754653
MPA	0.2057296	3.4704887	9.0366239	0.2057296	1.7248523
TSA	0.2041956	3.4953797	9.0641911	0.2061564	1.7338449
WOA	0.2137287	3.3297286	8.9738153	0.2209982	1.8213232
MVO	0.2059931	3.46481	9.044686	0.2060556	1.7283648
GWO	0.205592	3.4736454	9.0362401	0.2057988	1.7255236
TLBO	0.315253	4.4215666	6.7977001	0.4250886	3.0235653
GSA	0.2938352	2.7217307	7.4212433	0.3079408	2.0844573
PSO	0.3725304	3.4246823	7.3446854	0.5739303	4.0226813
GA	0.224308	6.9143503	7.7634846	0.304362	2.7608802

**Table 11 biomimetics-09-00137-t011:** Statistical results of optimization algorithms on the welded beam design problem.

Algorithm	Mean	Best	Worst	Std	Median	Rank
BOA	1.7246798	1.7246798	1.7246798	2.28E-16	1.7246798	1
WSO	1.7248526	1.7248523	1.7248578	1.25E-06	1.7248523	3
AVOA	1.7612095	1.7259197	1.8426669	0.0364639	1.7473196	7
RSA	2.1815628	1.9754653	2.5291486	0.1441236	2.1565285	8
MPA	1.7248523	1.7248523	1.7248523	3.35E-09	1.7248523	2
TSA	1.7431468	1.7338449	1.7523381	0.005605	1.743243	6
WOA	2.3106389	1.8213232	4.0458397	0.6416317	2.0857253	9
MVO	1.7412206	1.7283648	1.775042	0.0137552	1.7371509	5
GWO	1.7272522	1.7255236	1.7312956	0.0013626	1.727007	4
TLBO	3.326E+13	3.0235653	3.209E+14	8.111E+13	5.6909484	12
GSA	2.4436073	2.0844573	2.7521417	0.1914907	2.4731468	10
PSO	4.586E+13	4.0226813	2.776E+14	8.759E+13	6.7293081	13
GA	1.126E+13	2.7608802	1.218E+14	3.456E+13	5.6575496	11

**Table 12 biomimetics-09-00137-t012:** Performance of optimization algorithms on the tension/compression spring design problem.

Algorithm	Optimal Variables			Optimal Cost
	*d*	*D*	*P*	
BOA	0.0516891	0.3567177	11.288966	0.0126019
WSO	0.0516871	0.3566701	11.291759	0.0126652
AVOA	0.0511918	0.3448817	12.021301	0.0126702
RSA	0.0501316	0.314172	14.710881	0.0131579
MPA	0.0516907	0.3567583	11.28659	0.0126652
TSA	0.0509889	0.3401015	12.347641	0.012682
WOA	0.0511663	0.3442823	12.06054	0.0126707
MVO	0.0501316	0.3199667	13.884692	0.0127497
GWO	0.0519561	0.3631596	10.925567	0.0126707
TLBO	0.0677281	0.8916127	2.7236846	0.0174771
GSA	0.0551098	0.4411042	7.8215101	0.0130734
PSO	0.0676458	0.8885012	2.7236846	0.0173752
GA	0.0681952	0.8994084	2.7236846	0.0178708

**Table 13 biomimetics-09-00137-t013:** Statistical results of optimization algorithms on the tension/compression spring design problem.

Algorithm	Mean	Best	Worst	Std	Median	Rank
BOA	0.0126019	0.0126019	0.0126019	6.88E-18	0.0126019	1
WSO	0.0126763	0.0126652	0.0128239	3.537E-05	0.0126656	3
AVOA	0.0133339	0.0126702	0.0141329	0.00055	0.0132665	8
RSA	0.0132385	0.0131579	0.0133806	6.845E-05	0.0132178	6
MPA	0.0126652	0.0126652	0.0126652	2.81E-09	0.0126652	2
TSA	0.0129585	0.012682	0.0135147	0.0002383	0.0128858	5
WOA	0.0132643	0.0126707	0.0144745	0.0005961	0.0130687	7
MVO	0.0164236	0.0127497	0.0178419	0.0016251	0.0173272	9
GWO	0.0127222	0.0126707	0.0129425	5.456E-05	0.0127197	4
TLBO	0.0180015	0.0174771	0.0186004	0.0003532	0.0179579	10
GSA	0.0193335	0.0130734	0.031807	0.0042027	0.0189131	11
PSO	2.064E+13	0.0173752	3.663E+14	8.195E+13	0.0173752	13
GA	1.612E+12	0.0178708	1.668E+13	4.815E+12	0.025383	12

## Data Availability

Data are contained within the article.

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
