# Peer review of "Botox Optimization Algorithm: A New Human-Based Metaheuristic Algorithm for Solving Optimization Problems"

_biomimetics, 2024, doi:10.3390/biomimetics9030137_

Round 1

Reviewer 1 Report

Comments and Suggestions for Authors

1. The author proposes a novel algorithm, and should add Population diversity analysis and Exploration and Exploration analysis to this algorithm in the paper.

2. In the literature review section, the author should add an analysis of some advanced improved algorithms, such as:

https://doi.org/10.1016/j.eswa.2023.120594

https://doi.org/10.1016/j.asoc.2022.109777

https://doi.org/10.1016/j.eswa.2024.123444

3. The author should add a schematic diagram of the Botox injection, as Fig.1 does not help readers better understand.

4. In the results table, the author should bold the optimal value.

5. Why did the author choose these comparative algorithms and based on what?

6. In the Conclusions and Future Works section, the author will provide a detailed introduction to the motivation and characteristics of the BOA algorithm.

Author Response

Thank you for your expert opinion, and we hope that our changes will be sufficient for you.

Reviewer 2 Report

Comments and Suggestions for Authors

The article presents a novel metaheuristic algorithm, the Botox Optimization Algorithm (BOA), inspired by the principles of Botox injection procedures. The authors effectively outline the motivation behind BOA and its theoretical underpinnings, providing a clear understanding of how the algorithm operates. The rigorous evaluation of BOA's performance against the CEC 2017 test suite and its successful application to real-world optimization problems from the CEC 2011 test suite are commendable, demonstrating the algorithm's efficacy and versatility.

One of the strengths of the article is its structured approach, with sections dedicated to literature review, algorithm introduction, simulation studies, and real-world applications. This organization enhances readability and aids in comprehending the progression of the research.

However, there are a few areas that could be improved or clarified:

The article could benefit from a more detailed explanation of how BOA specifically draws inspiration from Botox procedures. While the analogy is intriguing, a deeper exploration of the parallels between the two could provide additional insights into the algorithm's design rationale.

Although the comparative analysis with twelve established metaheuristic algorithms highlights BOA's superiority, it would be valuable to include discussions on specific strengths and weaknesses compared to each algorithm. This would provide readers with a more nuanced understanding of BOA's performance and potential areas for improvement.

While the authors mention future research directions, such as exploring binary and multi-objective versions of BOA, more concrete suggestions or hypotheses for these avenues would enhance the conclusion section. Providing potential applications in diverse scientific domains could also stimulate further interest in the algorithm's potential impact.

Overall, the article presents a promising contribution to the field of metaheuristic optimization. By addressing the suggested improvements, the authors can further strengthen the clarity and impact of their research.

Author Response

(The authors gave the same response as above.)
